# Exploring Generative Neural Temporal Point Process

**Haitao Lin**                        *linhaitao@westlake.edu.cn*
*CAIRI, Westlake University*
*Zhejiang University*

**Lirong Wu**                        *wulirong@westlake.edu.cn*
*CAIRI, Westlake University*
*Zhejiang University*

**Guojiang Zhao**                     *guojianz@andrew.cmu.edu*
*CAIRI, Westlake University*
*Carnegie Mellon University*

**Pai Liu**                         *pailiu@westlake.edu.cn*
*School of Engineering, Westlake University*

**Stan Z. Li**                       *Stan.ZQ.Li@westlake.edu.cn*
*CAIRI, Westlake University*

**Reviewed on OpenReview:** *https://openreview.net/forum?id=NPfS5N3jbL*

## Abstract

Temporal point process (TPP) is commonly used to model the asynchronous event sequence featuring occurrence timestamps and revealed by probabilistic models conditioned on historical impacts. While lots of previous works have focused on 'goodness-of-fit' of TPP models by maximizing the likelihood, their predictive performance is unsatisfactory, which means the timestamps generated by models are far apart from true observations. Recently, deep generative models such as denoising diffusion and score matching models have achieved great progress in image generating tasks by demonstrating their capability of generating samples of high quality. However, there are no complete and unified works exploring and studying the potential of generative models in the context of event occurence modeling for TPP. In this work, we try to fill the gap by designing a unified **g**enerative framework for **n**eural **t**emporal **p**oint **p**rocess (GNTPP) model to explore their feasibility and effectiveness, and further improve models' predictive performance. Besides, in terms of measuring the historical impacts, we revise the attentive models which summarize influence from historical events with an adaptive reweighting term considering events' type relation and time intervals. Extensive experiments have been conducted to illustrate the improved predictive capability of GNTPP with a line of generative probabilistic decoders, and performance gain from the revised attention. To the best of our knowledge, this is the first work that adapts generative models in a complete unified framework and studies their effectiveness in the context of TPP. Our codebase including all the methods given in Section. 5.1.1 is open in `https://github.com/BIRD-TAO/GNTPP`. We hope the code framework can facilitate future research in Neural TPPs.

## 1 Introduction

Various forms of human activity or natural phenomena can be represented as discrete events happening with irregular time intervals, such as electronic health records, purchases in e-commerce systems, and earthquakes

with aftershocks. A natural choice for revealing the underlying mechanisms of the occurrence of events is temporal point processes (TPPs) (D.J. Daley, 2003; Isham & Westcott, 1979; Hawkes, 1971), which describe the probability distribution of time intervals and types of future events' occurrence by summarizing the impacts of historical observations.

Recently, with the rapid development in deep learning, TPP models also benefit from great expressiveness of neural networks, from the first work proposed in Du et al. (2016). A neural TPP model can be divided into two parts – **history encoder** and **probabilistic decoder**. Recently, great success has been achieved in modeling the TPPs thanks to fast developments in sequential models and generative models in deep learning (Shchur et al., 2021; Lin et al., 2021). The former concentrates on the competence of encoding and aggregating the past impacts of events on the next event's occurrence probability (Zhang et al., 2020a; Zuo et al., 2020), which is called history encoder; The latter one aims to improve the flexibility and efficiency to approximate the target distribution of occurrence time intervals conditioned on the historical encoding (Omi et al., 2019; Shchur et al., 2020a;b), which is called probabilistic decoder.

Most of the previous works focus on the 'goodness-of-fit' of the proposed models, which can be quantified by negative log-likelihood (NLL). However, limited by this, to model the distribution in a general fashion, one needs to formulate the probabilistic decoder with certain families of functions whose likelihoods are tractable, such as the mixture of distributions whose support sets are positive real numbers (Shchur et al., 2020a; Lin et al., 2021) or triangular maps as a generalization of autoregressive normalizing flows (Shchur et al., 2020b). Although some probabilistic decoders with intractable likelihoods still perform well in the evaluation of 'goodness-of-fit' (Mei & Eisner, 2017; Zhang et al., 2020a; Zuo et al., 2020), they rely on the stochastic or numerical approximation to calculate the likelihood, which leads to unaffordable high computational cost despite their theoretically universal approximation ability (Soen et al., 2021). To sum up, both the requirements for a certain structure in the functional approximators, e.g. on the mixture of log-normal distribution (Shchur et al., 2020a), and the excessive emphasis on 'goodness-of-fit' as the optimization objective considerably impose restrictions on the models' predictive and extrapolation performance. This causes that the timestamp samples generated by them are far apart from the ground-truth observations, which limits their application in real-world scenarios. Recent empirical studies show that these models' predictive performance is very unsatisfactory (Lin et al., 2021), with extremely high error in the task of next arrival time prediction. As a probabilistic models, a good TPP model should not only demonstrate its goodness-of-fitting (lower NLL), but also have ability to generate next arrival time as samples of high quality, as well as preserve the randomness and diversity of the generated samples.

Since the TPP models aim to approximate the target distribution of event occurrence conditioned on the historical influences, we can classify the TPP models into conditional probabilistic or generative models in the field of deep learning, and lend the techniques in these fields to improve the predictive performance (Yan et al., 2018; Xiao et al., 2017a). In deep probabilistic models, the functional forms in energy-based models are usually less restrictive, and the optimization of the objective function as unnormalized log-likelihood can be directly converted into a point estimation or regression problem, thus empowering the models to have an improved predictive ability. Recently, deep generative models including denoising diffusion models (Sohl-Dickstein et al., 2015b; Ho et al., 2020) and score matching models (Song et al., 2021b;a; Bao et al., 2022) as an instance of energy-based deep generative model have attracted lots of scientific interests as it demonstrates great ability to generate image samples of high quality. Thanks to its less restrictive functional forms and revised unnormalized log-probability as the optimization objective, in other fields such as crystal material generation (Xie et al., 2021) and time series forecasting (Rasul et al., 2021), they are also employed for generative tasks and demonstrates great feasibility. Enlighted by this, we conjecture that the probabilistic decoders constructed by deep generative models in the context of TPP are likely to generate samples of time intervals that are close to ground truth observations, thus improving the predictive performance. In this way, we design a unified framework for **g**enerative **n**eural **t**emporal **p**oint **p**rocess (GNTPP) by employing the deep generative models as the probabilistic decoder to approximate the target distribution of occurrence time. Besides, we revise the self-attentive history encoders (Zhang et al., 2020a) which measure the impacts of historical events with adaptive reweighting terms considering events' type relation and time intervals.

In summary, the contributions of this paper are listed as follows:

- To the best of our knowledge, we are the first to establish a complete framework of generative models to study their effectiveness for improving the predictive performance in TPPs, i.e. enabling TPP models to generate time sample of high quality.

- In terms of history encoders, we revise the self-attentive encoders with adaptive reweighting terms, considering type relation and time intervals of historical observations, showing better expressiveness.

- We conduct extensive experiments on one complicated synthetic dataset and four real-world datasets, to explore the potential of GNTPP in the aspect of predictive performance and fitting ability. Besides, further studies give more analysis to ensure the effectiveness of the revised encoder.

## 2 Background

### 2.1 Temporal Point Process

#### 2.1.1 Preliminaries

A realization of a TPP is a sequence of event timestamps $\{t_i\}_{1 \leq i \leq N}$, where $t_i \in \mathbb{R}^+$, and $t_i < t_{i+1} \leq T$. In a marked TPP, it allocates a type $m_i$ (a.k.a. mark) to each event timestamps, where there are $M$ types of events in total and $m_i \in [M]$ with $[M] = \{1, 2, \ldots, M\}$. A TPP is usually described as a counting process, with the measure $\mathcal{N}(t)$ defined as the number of events occurring in the time interval $(0, t]$.

We indicate with $\{(t_i, m_i)\}_{1 \leq i \leq N}$ as an observation of the process, and the history of a certain timestamp $t$ is denoted as $\mathcal{H}(t) = \{(t_j, m_j), t_j < t\}$. In this way, the TPP can be characterized via its intensity function conditioned on $\mathcal{H}(t)$, defined as

$$\lambda^*(t) = \quad \lambda(t|\mathcal{H}(t)) = \lim_{\Delta t \to 0^+} \frac{\mathbb{E}[\mathcal{N}(t + \Delta t) - \mathcal{N}(t)|\mathcal{H}(t)]}{\Delta t}, \tag{1}$$

which means the expected instantaneous rate of happening the events given the history. Note that it is always a non-negative function of $t$. Given the conditional intensity, the probability density function of the occurrence timestamps reads

$$q^*(t) = \lambda^*(t) \exp(-\int_{t_{i-1}}^{t} \lambda^*(\tau)d\tau), \tag{2}$$

The leading target of TPPs is to parameterize a model $p_\theta^*(t)$, to fit the distribution of the generated marked timestamps, i.e. $q^*(t)$, as to inference probability density or conditional intensity for further statistical prediction, such as next event arrival time prediction. Besides, in marked scenarios, parameterizing $p_\theta^*(m)$ to predict the next event type is also an important task. More details on preliminaries are given in Appendix A. Usually, in the deep neural TPP models, the impacts of historical events $\mathcal{H}(t)$ on the distribution of time $t$ are summarized as a historical encoding $\boldsymbol{h}_{i-1}$, where $i - 1 = \arg\max_{j \in \mathbb{N}} t_j < t$, and the parameters in $p_\theta^*(t)$ and $p_\theta^*(m)$ are determined by $\boldsymbol{h}_{i-1}$, for $t > t_i$, which reads

$$p^*(t; \theta(\boldsymbol{h}_{i-1})) = p_\theta(t|\boldsymbol{h}_{i-1}); \quad p^*(m; \theta(\boldsymbol{h}_{i-1})) = p_\theta(m|\boldsymbol{h}_{i-1}) \tag{3}$$

In summary, in deep neural TPPs, there are two key questions to answer in modeling the process:

(1) How to measure the historical events' impacts on the next events' arrival time and type distribution. In other words, how to encode the historical events before time $t$ which is $\mathcal{H}(t)$ for $t_{i-1} < t$ into a vector $\boldsymbol{h}_{i-1}$ to parameterize $p_\theta(t|\boldsymbol{h}_{i-1})$ or $p_\theta(m|\boldsymbol{h}_{i-1})$?

(2) How to use a conditional probabilistic model $p^*(t, m; \theta(\boldsymbol{h}_{i-1})) = p_\theta(t, m|\boldsymbol{h}_{i-1})$ with respect to time $t$ and type $m$, whose parameters are obtained by $\theta = \theta(\boldsymbol{h}_{i-1})$, to approximate the true probability of events' time and types?

### 2.1.2 History Encoders

For the task (1), it can be regarded as a task of sequence embedding, i.e. finding a mapping $H$ which maps a sequence of historical event time and types before $t$, i.e. $\mathcal{H}(t) = \{(t_j, m_j)\}_{1 \leq j \leq i-1}$ where $t > t_{i-1}$, to a vector $\boldsymbol{h}_{i-1} \in \mathbb{R}^D$ called historical encoding. $D$ is called '*embedding size*' in this paper.

To increase expressiveness, the $j$-th event in the history set is firstly lifted into a high-dimensional space, considering both temporal and type information, as

$$u(t_j, m_j) = \boldsymbol{e}_j = [\boldsymbol{\omega}(t_j); \boldsymbol{E}_m^T \boldsymbol{m}_j], \tag{4}$$

where $\boldsymbol{\omega}$ transforms one-dimension $t_j$ into a high-dimension vector, which can be linear, trigonometric, and so on, $\boldsymbol{E}_m$ is the embedding matrix for event types, and $\boldsymbol{m}_j$ is the one-hot encoding of event type $m_j$. Commonly, to normalize the timestamps into a unifying scale, the event embeddings take the $\tau_j = t_j - t_{j-1}$ as the inputs, i.e. $\boldsymbol{e}_j = u(\tau_j, m_j)$. And then, another mapping $v$ will be used to map the sequence of embedding $\{\boldsymbol{e}_1, \boldsymbol{e}_2, \ldots, \boldsymbol{e}_{i-1}\}$ into a vector space of dimension $D$, by

$$\boldsymbol{h}_{i-1} = v([\boldsymbol{e}_1; \boldsymbol{e}_2; \ldots; \boldsymbol{e}_{i-1}]). \tag{5}$$

For example, units of recurrent neural networks (RNNs) including GRU and LSTM can all be used to map the sequence (Du et al., 2016; Omi et al., 2019; Shchur et al., 2020a), as

$$\boldsymbol{h}_0 = \boldsymbol{0}; \quad \boldsymbol{h}_i = \text{RNN}(\boldsymbol{e}_i, \boldsymbol{h}_{i-1}) \tag{6}$$

Therefore, the history encoder can be dismantled as two parts, as

$$\begin{aligned} \boldsymbol{h}_{i-1} = H(\mathcal{H}(t)) &= v \circ u(\mathcal{H}(t)) \\ &= v([u(\tau_1, m_1); u(\tau_2, m_2); \ldots; u(\tau_{i-1}, m_{i-1})]), \end{aligned} \tag{7}$$

where $u$ and $v$ are the event encoder and sequence encoder respectively, and the composites of both make up the history encoder.

The attention mechanisms (Vaswani et al., 2017) which have made great progress in language models prove to be superior history encoders for TPPs in the recent research (Zhang et al., 2020a; Zuo et al., 2020). In this paper, we follow their works in implementing self-attention mechanisms (Vaswani et al., 2017) but conduct revisions to the self-attentive encoders, leading to our revised attentive history encoders in our paper.

### 2.1.3 Probabilistic Decoders

For the task (2), it is usually regarded as setting up a conditional probabilistic model $p^*(t, m|\theta(\boldsymbol{h}_{i-1}))$, whose conditional information is contained by model's parameters $\theta(\boldsymbol{h}_{i-1})$ obtained by historical encoding. Statistical inference and prediction can be conducted according to the model, such as generating new sequences of events, or using expectation of time $t$ to predict the next arrival time. Besides, for interpretability, the relation among different types of events such as Granger causality (Xu et al., 2016; Eichler et al., 2016) inferred by probabilistic models also arises research interests.

In the temporal domain, one choice is to directly formulate the conditional intensity function $\lambda_\theta^*(t)$ (Du et al., 2016), cumulative hazard function $\Lambda_\theta^*(t) = \int_0^t \lambda_\theta^*(\tau)d\tau$ (Omi et al., 2019) or probability density function $p_\theta^*(t)$ (Shchur et al., 2020a), and to minimize the negative log-likelihood as the optimization objective. For example, the loss of a single event's arrival time reads

$$l_i = -\log \lambda_\theta(t_i|\boldsymbol{h}_{i-1}) + \int_{t_{i-1}}^{t_i} \lambda_\theta(t|\boldsymbol{h}_{i-1})dt. \tag{8}$$

However, as demonstrated in Equation. (2) and (8), minimizing the negative likelihood requires the closed form of probability density function, which limits the flexibility of the models to approximate the true probabilistic distribution where the event data are generated. For example, one attempts to formulate $\lambda_\theta^*(t)$ will inevitably confront whether the integration of it (a.k.a. cumulative hazard function) has closed forms, for

Table 1: Description of exisiting neural TPP methods, following Lin et al. (2021).

| Methods | History Encoder | Probabilistic Decoder | Closed Likelihood | Flexible Sampling |
|---------|-----------------|-----------------------|-------------------|-------------------|
| RMTPP(Du et al., 2016) | RNN | Gompertz | ✔ | ✔ |
| LogNorm(Shchur et al., 2020a) | RNN | Log-normal | ✔ | ✔ |
| ERTPP(Xiao et al., 2017b) | RNN | Gaussian | ✔ | ✔ |
| WeibMix(Lin et al., 2021) | Transformer | Weibull | ✔ | ✔ |
| FNNInt(Omi et al., 2019) | RNN | Feed-forward Network | ✔ | ✗ |
| SAHP(Zhang et al., 2020a) | Transformer | Exp-decayed + Nonlinear | ✗ | ✗ |
| THP(Zuo et al., 2020) | Transformer | Linear-decayed + Nonlinear | ✗ | ✗ |
| WasGANTPP(Xiao et al., 2017a) | RNN | GAN | - | ✔ |

manageable computation of the likelihood. Although this term can be approximated by numerical or Monte Carlo integration methods, the deviation of the approximation from the analytical likelihood may occur due to insufficient sampling and high computational cost may be unaffordable. Another problem is that these likelihood-targeted models usually perform unsatisfactorily in next arrival time prediction (Lin et al., 2021) despite its good fitting capability in terms of negative log-likelihood.

For these reasons and enlightened by effectiveness of adversarial and reinforcement learning (Yan et al., 2018; Arjovsky et al., 2017; Upadhyay et al., 2018; Li et al., 2018) in the context of TPPs, we conjecture that state-of-the-art methods and techniques in deep generative models can be transferred to deep neural TPPs. Differing from the previous works focusing on models' fitting ability in terms of higher likelihood, these models aim to promote models' prediction ability, i.e. to generate high-quality samples which are closer to ground truth observations. Inspired by great success achieved by recently proposed generative models (Sohl-Dickstein et al., 2015a; Ho et al., 2020; Song et al., 2021b) which enjoy the advantages in generating image samples of good quality and have been extended to a line of fields (Xie et al., 2021; Rasul et al., 2021), we hope to deploy this new state-of-the-art probabilistic model to TPPs, to solve the dilemma of unsatisfactory predictive ability of neural TPPs as well as further enhance models' flexibility.

## 2.2  A Brief Review

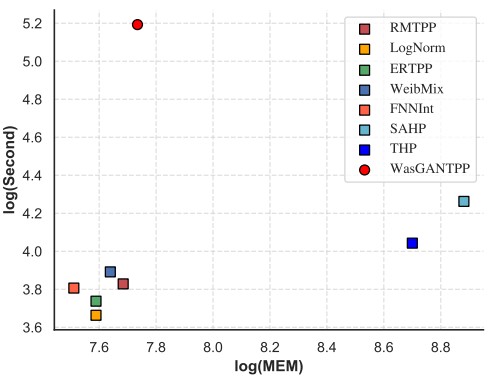

(a) Complexity Comparison on `MOOC`

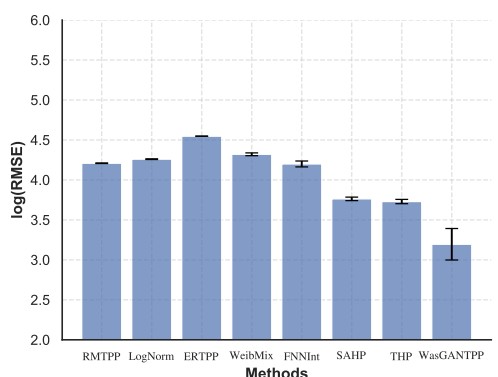

(b) Predictive Performance Comparison on `MOOC`

Figure 1: Intuitive explanation of our motivation: In (a), SAHP and THP cost more memory than others, and WasGANTPP is more time-consuming because of the adversarial training. In (b), the 'MAPE' in WasGANTPP as a generative model is relatively smaller, in comparison to the classical TPP probabilistic decoders. The detail of the experimental settings and results is given in Section 5 and Appendix B.4.

We review recently-proposed neural TPP models, and give a brief disciption of them in Table 1. Most methods directly model the intensity function or probabilistic density function of the process, while only WasGANTPP (Xiao et al., 2017a) employed Wasserstein GAN as the probabilistic decoder, whose learning target is an approximation to the Wasserstein distance between the empirical and model distributions, and allows flexible sample generation. The classical methods with closed-form likelihood usually show unsatisfactory performance,

while SAHP, THP and WASGANTPP achieve improvements, as shown in Figure. 1(b). SAHP and THP depends on numerical or Monte Carlo integration to approximate the likelihood because the probabilistic decoder has no closed form, leading to higher computational cost (shown in Figure 1(a)). Besides, FNNINT, SAHP and THP do not allow flexible sampling, which limits the real-world application when one needs to draw out samples from the learned models. Figure 1 adn Table 1 give an intuitive demonstration of our motivation.

## 3 Methodology

### 3.1 Revised Attentive Encoder

Self-attention as the key module in Transformer (Vaswani et al., 2017; Dong et al., 2021) benefits from its fast parallel computing and capability of encoding long-term sequences in lots of fields. In attentive TPPs (Zhang et al., 2020a), the events are embedded as vectors $\boldsymbol{e}_j = [\boldsymbol{\omega}(\tau_j); \boldsymbol{E}_m^T \boldsymbol{m}_j]$ by positional encoding techniques, where

$$\boldsymbol{\omega}(\tau_j) = [\sin(\omega_1 j + \omega_2 \tau_j); \cos(\omega_1 j + \omega_2 \tau_j)], \tag{9}$$

in which $\omega_1 j$ is positional term, and $\omega_2 \tau_j$ is time term. Or in Zuo et al. (2020), it reads

$$\boldsymbol{\omega}(\tau_j) = [\sin(\omega_1 \tau_j); \cos(\omega_2 \tau_j)]. \tag{10}$$

After that, the historical encoding obtained by attention mechanisms can be written as

$$\boldsymbol{h}_{i-1} = \sum_{j=1}^{i-1} \exp(\phi(\boldsymbol{e}_j, \boldsymbol{e}_{i-1})) \psi(\boldsymbol{e}_j) / \sum_{j=1}^{i-1} \exp(\phi(\boldsymbol{e}_j, \boldsymbol{e}_{i-1})), \tag{11}$$

where $\phi(\cdot, \cdot)$ maps two embedding into a scalar called attention weight, e.g. $\phi(\boldsymbol{e}_j, \boldsymbol{e}_i) = (\boldsymbol{e}_j W_Q)(\boldsymbol{e}_i W_K)^T$ and $\psi$ transforms $\boldsymbol{e}_j$ into a series of $D$-dimensional vectors called values, e.g. $\psi(\boldsymbol{e}_j) = \boldsymbol{e}_j W_V$. This calculation of Equation. (11) can be regarded as summarizing all the previous events' influence, with different weights $w_{j,i-1} = \frac{\exp(\phi(\boldsymbol{e}_j, \boldsymbol{e}_{i-1}))}{\sum_{j=1}^{i-1} \exp(\phi(\boldsymbol{e}_j, \boldsymbol{e}_{i-1}))} = \text{softmax}(\phi(\boldsymbol{e}_j, \boldsymbol{e}_{i-1}))$.

Although the self-attentive history encoders are very expressive and flexible for both long-term and local dependencies, which prove to be effective in deep neural TPPs (Zuo et al., 2020; Zhang et al., 2020a), we are motivated by the following two problems raised by event time intervals and types, and revise the classical attention mechanisms by multiplying two terms which consider time intervals and type relation respectively.

**P.1.** As shown in Equation. (11), in the scenarios where the positional encoding term $j$ is not used (Refer to Equation. (2) in Zuo et al. (2020)), if there are two events, with time intervals $\tau_{j_1} = t_{j_1} - t_{j_1-1}$ and $\tau_{j_2} = t_{j_2} - t_{j_2-1}$ which are equal and of the same type but $t_{j_2} > t_{j_1}$, the attention weights of them will be totally equal because their event embeddings $\boldsymbol{e}_{j_1}$ and $\boldsymbol{e}_{j_2}$ are the same. However, the impacts of the $j_2$-th and $j_1$-th event on time $t$ can be not necessarily the same, i.e. when the short-term events outweigh long-term ones, the impacts of $t_{j_2}$ should be greater, since $t_{j_2} > t_{j_1}$.

**P.2.** As discussed, the relations between event types are informative, which can provide interpretation of the fundamental mechanisms of the process. Self-attention can provide such relations, through averaging the attention weights of certain type of events to another (Zhang et al., 2020a). In comparison, we hope that attention weights are just used for expressiveness, and the relations among events should be provided by other modules.

To solve the problem **P.1.**, we revise the attention weight by a time-reweighting term by $\exp\{a(t - t_j)\}$, where $a$ is a learnable scaling parameter. This term will force the impacts of short-term events to be greater $(a < 0)$ or less $(a \geq 0)$ than ones of long-term events, when the two time intervals are the same. Although the position term in Zhang et al. (2020a) can also fix the problem, the exponential term can slightly improve the performance thanks to it further enhances models' expressivity (See Section 5.3).

To provide events' relations learned by the model as well as avoid flexibility of attention mechanisms as discussed in **P.2.**, we employ the type embedding $\boldsymbol{E}$ to calculate the cosine similarity of different types, and the inner product of two embedding vectors is used as another term to revise the attention weights (Zuo et al., 2020; Zhang et al., 2021). In this way, the weight of event $j$ in attention mechanisms can be written as

$$w_{j,i-1} = \text{softmax}((\boldsymbol{E}_m^T\boldsymbol{m}_j)^T(\boldsymbol{E}_m^T\boldsymbol{m}_{i-1})\exp\{a(t_{i-1}-t_j)\}\phi(\boldsymbol{e}_j,\boldsymbol{e}_{i-1})), \tag{12}$$

where $\boldsymbol{E}_m^T\boldsymbol{m}$ is normalized as a unit vector for all $m \in [M]$ as type embedding, and thus the inner product is equivalent to cosine similarity. Note that in Eq. 12, when $(\boldsymbol{E}_m^T\boldsymbol{m}_j)^T(\boldsymbol{E}_m^T\boldsymbol{m}_{i-1}) = 0$, the influence of $\boldsymbol{m}_j$ to $\boldsymbol{m}_{i-1}$ will not be eliminated after the softmax($\cdot$), so in implementation, we map $(\boldsymbol{E}_m^T\boldsymbol{m}_j)^T(\boldsymbol{E}_m^T\boldsymbol{m}_{i-1})$ to $-10^9$ to force the influence of type $\boldsymbol{m}_j$ events to $\boldsymbol{m}_{i-1}$ to be zero after softmax function if $(\boldsymbol{E}_m^T\boldsymbol{m}_j)^T(\boldsymbol{E}_m^T\boldsymbol{m}_{i-1}) = 0$. We deploy the revised attentive encoder into the Transformer, which are commonly used in Zhang et al. (2020a); Zuo et al. (2020).

## 3.2 Generative Probabilistic Decoder

In the generative model, we directly model the time intervals instead of timestamps. For the observation $\{t_j\}_{j \leq i-1}$, the next observed arrival time interval is $\tau_i = t_i - t_{i-1}$, while the next sampled arrival time is $\hat{t}_i = \hat{\tau}_i + t_{i-1}$ after the sample $\hat{\tau}_i$ is obtained.

### 3.2.1 Temporal Conditional Diffusion Denoising Probabilistic Model

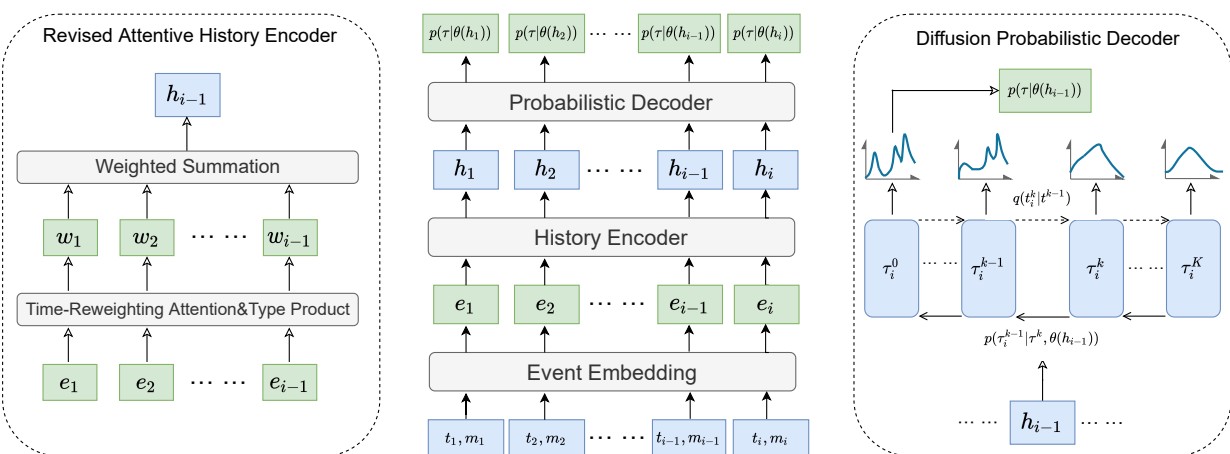

Figure 2: The workflows of revised attentive history encoder with TCDDM as probabilistic decoder.

**Temporal Diffusion Denoising Probabilistic Decoder.** For notation simplicity, we first introduce the temporal diffusion denoising decoder with no historical encoding as condition, and the $i$-th sample $\tau_i$ is denoted by $\tau$ in that we are discussing occurrence of a single event. Denote the a single observation of event occurrence time interval by $\tau = \tau^0 \sim q(\tau^0)$, where $\tau^0 \in \mathbb{R}^+$ and $q(\tau^0)$ is unkown, and the probability density function by $p_\theta(\tau^0)$ which aims to approximate $q(\tau^0)$ and allows for easy sampling. Diffusion models (Sohl-Dickstein et al., 2015a) are employed as latent variable models of the form $p_\theta(\tau^0) := \int p_\theta(\tau^{0:K})\,\mathrm{d}\tau^{1:K}$, where $\tau^1, \ldots, \tau^K$ are latent variables. The approximate posterior $q(\tau^{1:K}|\tau^0)$,

$$q(\tau^{1:K}|\tau^0) = \Pi_{k=1}^K q(\tau^k|\tau^{k-1}); \quad q(\tau^k|\tau^{k-1}) := \mathcal{N}(\tau^k; \sqrt{1-\beta_k}\tau^{k-1}, \beta_k). \tag{13}$$

is fixed to a Markov chain, which is called the 'forward process'. $\beta_1, \ldots, \beta_K \in (0,1)$ are predefined parameters. 'Reverse process' is also a Markov chain with learned Gaussian transitions starting with $p(\tau^K) = \mathcal{N}(\tau^K; 0, 1)$

$$p_\theta(\tau^{k-1}|\tau^k) := \mathcal{N}(\tau^{k-1}; \mu_\theta(\tau^k, k), \Sigma_\theta(\tau^k, k)), \tag{14}$$

The likelihood is not tractable, so the parameters $\theta$ are learned to fit the data distribution by minimizing the negative log-likelihood via its variational bound (Ho et al., 2020):

$$\min_\theta \mathbb{E}_{q(\tau^0)}[-\log p_\theta(\tau^0)] \leq \mathbb{E}_q \left[ \frac{1}{2\Sigma_\theta} \|\tilde{\mu}_k(\tau^k, \tau^0) - \mu_\theta(\tau^k, k)\|^2 \right] + C, \tag{15}$$

where $C$ is a constant which does not depend on $\theta$, and $\tilde{\mu}_k(\tau^k, \tau^0) := \frac{\sqrt{\bar{\alpha}_{k-1}}\beta_k}{1-\bar{\alpha}_k}\tau^0 + \frac{\sqrt{\bar{\alpha}_k}(1-\bar{\alpha}_{k-1})}{1-\bar{\alpha}_k}\tau^k$; $\tilde{\beta}_k := \frac{1-\bar{\alpha}_{k-1}}{1-\bar{\alpha}_k}\beta_k$. The optimization objective in Equation. (15) is straightforward since it tries to use $\mu_\theta$ to predict $\tilde{\mu}_k$ for every step $k$. To resemble learning process in multiple noise scales score matching (Song & Ermon, 2019; 2020), it further reparameterizing Equation. (15) as

$$\mathbb{E}_{\tau^0,\epsilon} \left[ \frac{\beta_k^2}{2\Sigma_\theta \alpha_k(1-\bar{\alpha}_k)} \|\epsilon - \epsilon_\theta(\sqrt{\bar{\alpha}_k}\tau^0 + \sqrt{1-\bar{\alpha}_k}\epsilon, k)\|^2 \right]. \tag{16}$$

since $\tau^k(\tau^0, \epsilon) = \sqrt{\bar{\alpha}_k}\tau^0 + \sqrt{1-\bar{\alpha}_k}\epsilon$ for $\epsilon \sim \mathcal{N}(0, 1)$ .

**Temporal Conditional Diffusion Denoising Probabilistic Decoder.** We establish a temporal conditional diffusion denoising model (TCDDM) as the probabilistic decoder in GNTPP. In training, after $\boldsymbol{h}_{i-1}$ is obtained as historical encoding, through a similar derivation in the previous paragraph, we can obtain the temporal conditional variant of the objective of a single event arrival time in Equation. (16) as

$$l_i = \mathbb{E}_{\tau_i^0,\epsilon} \left[ \|\epsilon - \epsilon_\theta(\sqrt{\bar{\alpha}_k}\tau_i^0 + \sqrt{1-\bar{\alpha}_k}\epsilon, \boldsymbol{h}_{i-1}, k)\|^2 \right], \tag{17}$$

in which the technique of reweighting different noise term is employed. $\epsilon_\theta$ as a neural network is conditioned on the historical encodings $\boldsymbol{h}_{i-1}$ and the diffusion step $k$. Our formutaion of $\epsilon_\theta$ is a feed-forward neural network, as

$$\begin{aligned} &\epsilon_\theta(\sqrt{\bar{\alpha}_k}\tau_i^0 + \sqrt{1-\bar{\alpha}_k}\epsilon, \boldsymbol{h}_{i-1}, k) \\ =\ &\boldsymbol{W}^{(3)}(\boldsymbol{W}^{(2)}(\boldsymbol{W}_h^{(1)}\boldsymbol{h}_{i-1} + \boldsymbol{W}_t^{(1)}\tau_i' + \cos(\boldsymbol{E}_k\boldsymbol{k})) + b^{(2)}) + b^{(3)}, \end{aligned} \tag{18}$$

where $\boldsymbol{W}_h^{(1)} \in \mathbb{R}^{D\times D}$, $\boldsymbol{W}_t^{(1)} \in \mathbb{R}^{D\times 1}$, $\boldsymbol{W}^{(2)} \in \mathbb{R}^{D\times D}$, $\boldsymbol{W}^{(3)} \in \mathbb{R}^{1\times D}$, $\tau_i' = \sqrt{\bar{\alpha}_k}\tau_i^0 + \sqrt{1-\bar{\alpha}_k}\epsilon$, $\boldsymbol{E}_k$ is the learnable embedding matrix of step $k$ and $\boldsymbol{k}$ is the one-hot encoding of $k$. In implementation, the residual block is used for fast and stable convergence.

In sampling, given the historical encoding $\boldsymbol{h}_{i-1}$, we first sample $\hat{\tau}_i^K$ from the standard normal distribution $\mathcal{N}(0, 1)$, then take it and $\boldsymbol{h}_{i-1}$ as the input of network $\epsilon_\theta$ to get the approximated noise, and generally remove the noise with different scales to recover the samples. This process is very similar to annealed Langevin dynamics in score matching methods.

For inference, the prediction is based on Monte Carlo estimation. For example, when mean estimation is deployed to predict the next event arrival time interval after $t_{i-1}$, we first sample a large amount of time interval from $p_\theta(\tau|\boldsymbol{h}_{i-1})$ (e.g. 100 times), then use the average of sampled $\{\hat{\tau}^{(s)}\}_{1\leq s\leq S}$ to estimate the mean of learned distribution, as the prediction value of next event arrival time interval, so the next arrival time is estimated as $\mathbb{E}[t_i] \approx t_{i-1} + \frac{1}{N}\sum_{s=1}^S \hat{\tau}^{(s)}$.

| **Algorithm 1** Training for each timestamp $t_i > t_{i-1}$ in temporal point process in TCDDM | **Algorithm 2** Sampling $\hat{t}_i > t_{i-1}$ via Langevin dynamics |
|---|---|
| 1: **Input:** Observation time interval $\tau_i$ and historical encoding $\boldsymbol{h}_{i-1}$ 
 2: **repeat** 
 3:     Initialize $k \sim \text{Uniform}(1,\ldots,K)$ and $\epsilon \sim \mathcal{N}(0,1)$ 
 4:     Take gradient step on 

      $\nabla_\theta\|\epsilon - \epsilon_\theta(\sqrt{\bar{\alpha}_k}\tau_i + \sqrt{1-\bar{\alpha}_k}\epsilon, \mathbf{h}_{i-1}, k)\|^2$ 

 5: **until** converged | **Input:** noise $\hat{\tau}_i^K \sim \mathcal{N}(0,1)$ and historical encoding $\mathbf{h}_{i-1}$ 
 **for** $k = K$ **to** 1 **do** 
    **if** $k > 1$ **then** 
      $z \sim \mathcal{N}(0,1)$ 
    **else** 
      $z = 0$ 
    **end if** 
    $\hat{\tau}_i^{k-1} = \frac{1}{\sqrt{\alpha_k}}(\hat{\tau}_i^k - \frac{\beta_k}{\sqrt{1-\bar{\alpha}_k}}\epsilon_\theta(\hat{\tau}_i^k, \mathbf{h}_{i-1}, k)) + \sqrt{\Sigma_\theta}z$ 
 **end for** 
 **Return:** $\hat{t}_i = \hat{\tau}_i^0 + t_{i-1}$ |

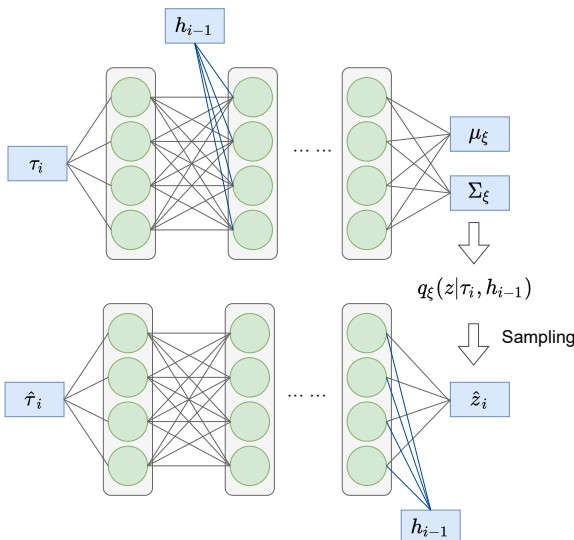

Figure 3: Network structure of temporal conditional variational autoencoder as the probabilistic decoder.

### 3.3 Temporal Conditional Variational AutoEncoder Probabilistic Model

We establish a temporal conditional variational autoencoder (TCVAE) as the probabilistic decoder (Kingma & Welling, 2014; Pan et al., 2020), which consists of a variational encoder $q_\xi(\boldsymbol{z}|\tau_i, \boldsymbol{h}_{i-1})$ as a conditional Gaussian distribution $\mathcal{N}(\mu_\xi, \Sigma_\xi)$ for approximating the prior standard Gaussian $\mathcal{N}(\mathbf{0}, \mathbf{I})$ and a variational decoder $p_\theta(\tau|\boldsymbol{z}_i, \boldsymbol{h}_{i-1})$ to generate arrival time samples. The network structure is given in Figure 3, where the latent Gaussian variable $\boldsymbol{z} \in \mathbb{R}^D$. The training objective of a single event's arrival time interval is the evidence lower bound, which can be written as

$$\min_{\theta,\xi} D_{\mathrm{KL}}(q_\xi(\boldsymbol{z}|\tau_i, \boldsymbol{h}_{i-1})|\mathcal{N}(\mathbf{0}, \mathbf{I})) + \mathbb{E}_{\hat{\tau}_i \sim p_\theta} \left[ \|\hat{\tau}_i - \tau_i\|_2^2 \right]. \tag{19}$$

In sampling process, the encoder $q_\xi(\boldsymbol{z}|\tau_i, \boldsymbol{h}_{i-1})$ is abandoned. The decoder $p_\theta(\tau|\boldsymbol{z}_i, \boldsymbol{h}_{i-1})$ transforms sample $\boldsymbol{z}_i$ which is generated from $\mathcal{N}(\mathbf{0}, \mathbf{I})$ to the target sample $\hat{\tau}_i$ conditioned on $\boldsymbol{h}_{i-1}$.

#### 3.3.1 Temporal Conditional Generative Adversarial Network Probabilistic Model

Our temporal conditional generative adversarial network (TCGAN) decoder is mostly based on WASSERSTEIN GAN in TPPs (Arjovsky et al., 2017; Xiao et al., 2017a). The probabilistic generator $p_\theta(\tau|\boldsymbol{z}, \boldsymbol{h}_{i-1})$ is trained via adversarial process, in which the other network called discriminator $d_\xi(\tau|\boldsymbol{h}_{i-1})$ is trained to map the samples to a scalar, for maximizing the Wasserstein distance between the distribution of generated samples $\hat{\tau}_i$ and the distribution of observed samples $\tau_i$. The final objective to optimize in TCGAN is

$$\min_\theta \max_\xi \mathbb{E}_{\hat{\tau}_i \sim p_\theta(\tau|\boldsymbol{z}, \boldsymbol{h}_{i-1})} \left[ d_\xi(\tau_i|\boldsymbol{h}_{i-1}) - d_\xi(\hat{\tau}_i|\boldsymbol{h}_{i-1}) \right] - \eta \left| \frac{|d_\xi(\tau_i|\boldsymbol{h}_{i-1}) - d_\xi(\hat{\tau}_i|\boldsymbol{h}_{i-1})|}{|\hat{\tau}_i - \tau_i|} - 1 \right|, \tag{20}$$

where the first term is to maximize the distance, and the second is to add a Lipschitz constraint as a regularization term proposed in original WASSERSTEIN GAN (Arjovsky et al., 2017) with $\eta$ as the loss weight. The formulation of the $p_\theta(\tau|\boldsymbol{z}, \boldsymbol{h}_{i-1})$ and $d_\xi(\tau|\boldsymbol{h}_{i-1})$ are similar to the variational decoder in the TCVAE, both transforming the $D$-dimensional random variables into 1. After training, $p_\theta(\tau|\boldsymbol{z}, \boldsymbol{h}_{i-1})$ can be used for sampling in the same process as the variational decoder in TCVAE.

#### 3.3.2 Temporal Conditional Continuous Normalizing Flow Probabilistic Model

As a classical generative model, normalizing flows (Papamakarios et al., 2019) are constructed by a series of invertible equi-dimensional mapping. However, in TPPs, the input data sample is 1-dimensional time, and

thus the flexibililty and powerful expressiveness of neural network is hard to harness. Therefore, here we choose to use temporal conditional continuous normalizing flows (TCCNF) (Mehrasa et al., 2020) which is based on Neural ODE (Chen et al., 2019; 2021). Note that the $t$ term in Neural ODE is here replaced by $k$, to avoid confusion. The TCCNF defines the distribution through the following dynamics system:

$$\tau_i = F_\theta(\tau(k_0)|\boldsymbol{h}_{i-1}) = \tau(k_0) + \int_{k_0}^{k_1} f_\theta(\tau(k), k|\boldsymbol{h}_{i-1})dk, \tag{21}$$

where $\tau(k_0) \sim \mathcal{N}(0, 1)$. $f_\theta$ is implemented with the same structure of variational decoder in TCVAE. The invertibility of $F_\theta(\tau(k_0)|\boldsymbol{h}_{i-1})$ allows us to not only conduct fast sampling, but also easily optimize the parameter set $\theta$ by minimizing the negative log-likelihood on a single time sample:

$$\min_\theta \left\{ -\log(p(\tau(k_0))) + \int_{k_0}^{k_1} \mathrm{Tr}(\frac{\partial f_\theta(\tau(k), k|\boldsymbol{h}_{i-1})}{\partial \tau(k)})dk|_{\tau=\tau_i} \right\}. \tag{22}$$

### 3.3.3 Temporal Conditional Noise Score Network Probabilistic Model

Finally, we establish the probabilistic decoder via a temporal conditional noise score network (TCNSN) as a score matching method, which aims to learn the gradient field of the target distribution (Song & Ermon, 2019; 2020). In specific, given a sequence of noise levels $\{\sigma_k\}_{k=1}^K$ with noise distribution $q_{\sigma_i}(\tilde{\tau}_i|\tau_i, \boldsymbol{h}_{i-1})$, i.e. $\mathcal{N}(\tilde{\tau}_i|\tau_i, \boldsymbol{h}_{i-1}, \sigma_k^2)$, the training loss for a single arrival time on each noise level $\sigma_k$ is as follows

$$l_i = \frac{1}{2}\|s_\theta(\tilde{\tau}_i; \sigma_k|\boldsymbol{h}_{i-1}) - \nabla_{\tilde{\tau}_i} \log q_{\sigma_k}(\tilde{\tau}_i|\tau_i, \boldsymbol{h}_{i-1})\|_2^2, \tag{23}$$

where the $s_\theta$ is the gradient of target distribution with the same formulation of variational decoders in TCVAE. According to Song & Ermon (2019), the weighted training objective can be written as

$$\min_\theta \frac{\sigma_k^2}{2}\|\frac{s_\theta(\tilde{\tau}_i; \sigma_k|\boldsymbol{h}_{i-1})}{\sigma_k} + \frac{\tilde{\tau}_i - \tau_i}{\sigma_k^2}\|_2^2, \tag{24}$$

where $\tilde{\tau}_i \sim q_{\sigma_k}(\tilde{\tau}_i|\tau_i, \boldsymbol{h}_{i-1})$.

In the sampling process, the Langevin dynamics (Welling & Teh, 2011) is used, in which the sample is firstly drawn from a Gaussian distribution, then iteratively updated by

$$\hat{\tau}_i^k = \hat{\tau}_i^{k-1} + \alpha_k s_\theta(\hat{\tau}_i^{k-1}; \sigma_k|\boldsymbol{h}_{i-1}) + \sqrt{2\alpha_k}z, \tag{25}$$

in different noise levels with different times, where $z \sim \mathcal{N}(0, 1)$.

### 3.4 Mark Modeling

When there exists more than one event type, another predictive target is what type of event is most likely to happen, given the historical observations. The task is regarded as a categorical classification. Based on the assumption that the mark and time distributions are conditionally independent given the historical embedding Shchur et al. (2021); Lin et al. (2021), we first transform the historical encoding $\boldsymbol{h}_{i-1}$ to the discrete distribution's logit scores as

$$\kappa(\boldsymbol{h}_{i-1}) = \mathrm{logit}(\hat{m}_i), \tag{26}$$

where $\mathrm{logit}(\hat{m}_i) \in \mathbb{R}^M$, $\kappa : \mathbb{R}^D \to \mathbb{R}^M$. Then, *softmax* function is used to transform logit scores into the categorical distribution, as

$$\mathrm{p}(\hat{m}_i = m|\theta(\boldsymbol{h}_{i-1})) = \mathrm{softmax}(\mathrm{logit}(\hat{m}_i))_m \tag{27}$$

where $\mathrm{softmax}(\mathrm{logit}(\hat{m}_i))_m$ means choose the $m$-th element after *softmax*'s output. In training, a cross-entropy loss for categorical classification $CE_i = \mathrm{CE}(p(m_i|\boldsymbol{h}_{i-1}))$ will be added to the loss term, leading the final loss of a single event to

$$L_i = l_i + \mathrm{CE}_i. \tag{28}$$

## 4   Related Work

**Deep Neural Temporal Point Process.**   From Du et al. (2016) which firstly employed RNNs as history encoders with a variant of Gompertz distribution as its probabilistic decoder. Following works, proposed to combine deep neural networks with TPPs, have achieved great progress (Lin et al., 2021; Shchur et al., 2021). For example, in terms of history encoder, a continuous time model (Mei & Eisner, 2017) used recurrent units and introduces a temporal continuous memory cell in it. Recently, attention-based encoder (Zhang et al., 2020a; Zuo et al., 2020) is established as a new state-of-the-art history encoder. In probabilistic decoders, Omi et al. (2019) fit the cumulative harzard function with its derivative as intensity function. Xiao et al. (2017b) and Shchur et al. (2020a) used the single Gaussian and the mixture of log-normal to approximate the target distribution respectively. In events dependency discovering, Mei et al. (2022) explicitly modeled dependencies between event types in the self-attention layer, and Zhang et al. (2020b) implicitly captured the underlying event interdependency by fitting a neural point process.

**Probabilistic Generative TPP Models.**   A line of works have been proposed to deploy new progress in deep generative models to TPPs. For example, the reinforcement learning approaches which are similar to adversarial settings, used two networks with one generating samples and the other giving rewards are proposed sequentially (Upadhyay et al., 2018; Li et al., 2018). And adversarial and discriminative learning (Yan et al., 2018; Xiao et al., 2017a) have been proposed to further improve the predictive abilities of probabilistic decoders. Noise contrastive learning to maximize the difference of probabilistic distribution between random noise and true samples also proved to be effective in learning TPPs (Guo et al., 2018; Mei et al., 2020).

## 5   Experiments

### 5.1   Experimental Setup

#### 5.1.1   Implementation Description

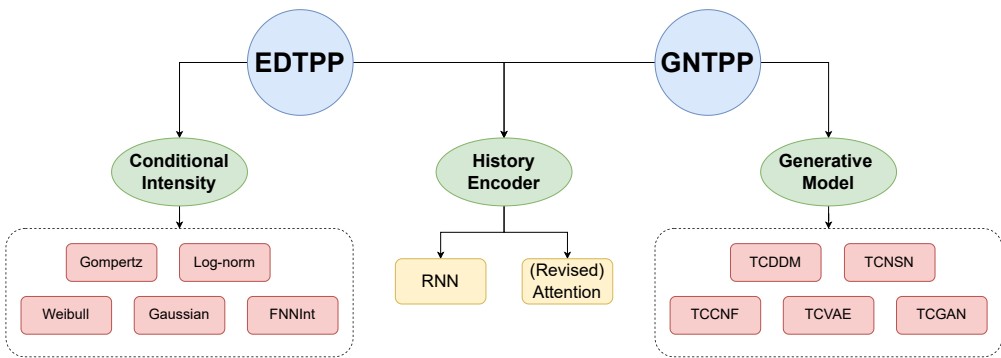

Figure 4: The hierarchical description of our experimental framework with modules integrated in GNTPP.

We first introduce our experimental framework for model comparison, as shown in Figure 4. In the history encoder module, it includes: GRU, LSTM and Transformer with and without our revised attention. In the probabilistic decoder module, probabilistic models in EDTPP (Lin et al., 2021) with closed likelihood including Gaussian, Gompertz, Log-norm, Feed-forward Network, and Weibull are implemented and integrated into our code. And the discussed neural generative models which is classified into our GNTPP including TCDDM, TCVAE, TCGAN, TCCNF and TCNSN are implemented.

#### 5.1.2   Datasets

We use a complex synthetic dataset which is simulated by Hawkes process of five types of events with different impact functions (Appendix B.1.) and 4 real-world datasets containing event data from various domains: MOOC (user interaction with online course system), Retweet (posts in social media), Stack Overflow

(question-answering badges in website), `Yelp` (check-ins to restaurants). The data statistics are shown in Table 2. We clamp the maximum length of sequences to 1000. The dataset is split into 20% ratio for testing, 80% ratio for training with 20% in training set as validation set used for parameter tuning. All the time scale $[0, T_{\max}]$ is normalized into $[0, 50]$ for numerical stability, where $T_{\max}$ is the maximum of observed event occurrence time in the training set. The detailed description is given in Appendix B.1.

Table 2: Dataset Statistics

| Dataset | # of sequences | Mean length | Min length | Max length | # of event type |
|---|---|---|---|---|---|
| MOOC | 7047 | 56.28 | 4 | 493 | 97 |
| Retweet | 24000 | 108.75 | 50 | 264 | 3 |
| Stack Overflow | 6633 | 72.42 | 41 | 736 | 22 |
| Yelp | 300 | 717.15 | 424 | 2868 | 1 |
| Synthetic | 6000 | 580.36 | 380 | 835 | 5 |

### 5.1.3 Protocol

In the training process, hyper-parameters of every model are tuned in the range of *'learning rate'*: $\{1 \times 10^{-3}, 5 \times 10^{-4}, 1 \times 10^{-4}\}$, *'embedding size'*: $\{8, 16, 32\}$, *'layer number'*: $\{1, 2, 3\}$, where '*embedding size*' is the dimension of historical encoding, i.e. $D$. The hyper-parameters are tuned on validation set. The maximum training epoch number is set as 100, and early stopping technique is used based on values of loss on validation set. The reported metrics are the results of models trained with the lowest loss, except 'TCGAN' probabilistic decoder, whose parameters are choosen as the final epoch's results. The mean and standard deviation of each metric is obtained according to 5 experiments' results with different random seeds.

### 5.1.4 Metrics

To evaluate the predictive performance of each methods, we deploy commonly used metric – 'mean absolute percent error' (MAPE) for measuring the predictive performance of next arrival time (Zhang et al., 2020a), and 'top-1 accuracy' (Top1_ACC) and 'top-3 accuracy' (Top3_ACC) to measure the predictive performance of next event types Lin et al. (2021). Note that there is only one event type in `Yelp`, so 'Top3_ACC' is not meaningful. However, the commonly used negative likelihood has no closed form in deep generative models. Therefore, we use another two metrics to evaluate the 'goodness-of-fitness'. The first is 'continuous ranked probability score' (CRPS), which is widely used in time series prediction (Rasul et al., 2021; Ben Taieb, 2022) for measuring the compatibility of a cumulative distribution function (CDF) $F$ with an observation $t$ as $\text{CRPS}(F, t) = \int_{\mathbb{R}} (F(y) - \mathbb{I}\{t \leq y\})^2 \, \mathrm{d}y$. Regarding the model as fitting next event arrival time's distribution (Jordan et al., 2018), we can calculate it on a single timestamp by using the empirical CDF as

$$\text{CRPS}(\hat{F}, t_i) = \frac{1}{S} \sum_{i=1}^{S} |\hat{t}_{i,k} - t_i| - \frac{1}{2S^2} \sum_{i=1}^{S} \sum_{j=1}^{S} |\hat{t}_{i,k} - \hat{t}_{j,k}|, \tag{29}$$

where there are $S$ samples $\{\hat{t}_{i,k}\}_{1 \leq k \leq S}$ drawn from $p_\theta(t|\boldsymbol{h}_{i-1})$, $t_i$ is the ground truth observation. Equation. (29) reflects that CRPS can also evaluate models' the sampling quality as predictive performance in the first term, and the sampling diversity in the second term. Another metric is 'QQPlot-deviation' (QQP-Dev) (Xiao et al., 2017a), which can be calculated by first computing the empirical cumulative hazard function $\hat{\Lambda}_\theta^*(t)$, and its distribution should be exponential with parameter 1. Therefore, the deviation of the QQ plot of it v.s. Exp(1) is calculated, as metric 'QQP-Dev'. Appendix B.2. gives details.

### 5.2 Performance Comparison

Here we choose 5 methods whose probabilistic decoders are not generative models as baseline for performance comparison:

*(1)* RMTPP (Lin et al., 2021) as the extension of RMTPP (Du et al., 2016), whose probabilistic decoder is a mixture of GOMPERTZ distribution.

*(2)* LogNorm (Shchur et al., 2020a), which uses Log-normal distribution as its decoder.

*(3)* ERTPP (Lin et al., 2021) as the generalization of ERTPP (Xiao et al., 2017b), with a mixture of Gaussian as its decoder.

*(4)* FNNInt (Omi et al., 2019), which formulates the cumulative harzard function as a fully-connected feed-forward network, and its derivative w.r.t. time as intensity.

*(5)* WeibMix (Marín et al., 2005; Lin et al., 2021) with Weibull mixture distribution as its decoder.

*(6)* SAHP (Zhang et al., 2020a), whose conditional intensity function is an exponential-decayed formulation, with a `softplus` as a nonlinear activation function stacked in the final.

*(7)* THP (Zuo et al., 2020), whose intensity function reads $\lambda^*(\tau) = \text{softplus}(\alpha \frac{\tau}{t_{i-1}} + \boldsymbol{W}_h \boldsymbol{h}_{i-1} + b)$.

*(8)* Deter, as a baseline for MAPE and ACC metrics, which is a totally deterministic model with the probabilistic model replaced by a linear projection head whose weight and bias are all constraint to be positive, and trained by MSE loss as a regression task.

Table 3: Comparison of different methods' performance on the real-world datasets. Results in **bold** give the top-3 performance, where Deter is excluded. The comparison on NLL or ELBO is given in Appendix B.3.

| | MOOC | | | | | Retweet | | | | |
|---|---|---|---|---|---|---|---|---|---|---|
| Methods | MAPE(↓) | CRPS(↓) | QQP_Dev(↓) | Top1_ACC(↑) | Top3_ACC(↑) | MAPE(↓) | CRPS(↓) | QQP_Dev(↓) | Top1_ACC(↑) | Top3_ACC(↑) |
| Deter | 17.4356±6.4756 | - | - | 0.3894±0.6027 | 0.70445±0.3361 | 12.7697±1.1566 | - | - | 0.5745±0.0001 | 1.0000±0.0000 |
| RMTPP | 67.2866±0.2321 | 37.1259±0.2539 | 1.9677±0.0002 | 0.4069±0.0130 | 0.7189±0.0131 | 65.1189±1.2747 | 0.3282±0.0075 | 1.7006±0.0035 | **0.6086**±0.0001 | 1.0000±0.0000 |
| LogNorm | 70.8006±0.3010 | 36.2675±0.6712 | 1.9678±0.0006 | 0.3992±0.0012 | 0.7155±0.0011 | 75.3065±0.0000 | 0.4579±0.0803 | 1.7091±0.0101 | 0.6003±0.0063 | 1.0000±0.0000 |
| ERTPP | 94.3711±0.0713 | 24.4113±0.3728 | 1.9571±0.0006 | 0.3841±0.0189 | 0.7043±0.0165 | 71.5601±0.0000 | 0.3842±0.0144 | 1.7283±0.0033 | 0.6055±0.0042 | 1.0000±0.0000 |
| WeibMix | 75.2570±1.3158 | 18.1352±4.0137 | 1.9776±0.0000 | 0.3409±0.0293 | 0.6613±0.0295 | 72.5045±0.4957 | 0.3795±0.0043 | 1.9776±0.0001 | 0.6058±0.0005 | 1.0000±0.0000 |
| FNNInt | 66.5765±2.4615 | - | 1.3780±0.0067 | **0.4203**±0.0035 | **0.7310**±0.0024 | 22.7489±3.8260 | - | 1.0318±0.0749 | 0.5348±0.0367 | 1.0000±0.0000 |
| SAHP | 43.0847±0.9447 | - | **1.0336**±0.0082 | 0.3307±0.0138 | 0.6527±0.0138 | 15.5689±0.0239 | - | **1.0286**±0.0030 | 0.6032±0.0001 | 1.0000±0.0000 |
| THP | 41.6676±1.1192 | - | **1.0207**±0.0001 | 0.3287±0.0097 | 0.6531±0.0109 | 16.4464±0.0112 | - | **1.0242**±0.0014 | 0.5651±0.0003 | 1.0000±0.0000 |
| TCDDM | **23.5559**±0.3098 | **0.1468**±0.0000 | 1.0369±0.0000 | **0.4308**±0.0069 | **0.7408**±0.0044 | **12.6058**±0.5550 | **0.2076**±0.0000 | **1.0327**±0.0111 | **0.6274**±0.0075 | 1.0000±0.0000 |
| TCVAE | **19.3336**±1.4021 | **0.1465**±0.0003 | **1.0369**±0.0001 | 0.3177±0.0066 | 0.6282±0.0032 | **12.2332**±0.6755 | **0.1848**±0.0005 | 1.0443±0.0018 | 0.5825±0.0213 | 1.0000±0.0000 |
| TCGAN | **24.4184**±4.7497 | **0.1470**±0.0001 | 1.0352±0.0001 | 0.4179±0.0049 | 0.7270±0.0005 | 15.4630±1.5843 | 0.2084±0.0134 | 1.0356±0.0002 | **0.6263**±0.0088 | 1.0000±0.0000 |
| TCCNF | 26.3197±1.7119 | 0.1636±0.0044 | 1.0578±0.0106 | **0.4297**±0.0105 | **0.7374**±0.0100 | **13.9865**±1.9811 | **0.1625**±0.0092 | 1.0598±0.0022 | 0.5965±0.0105 | 1.0000±0.0000 |
| TCNSN | 80.8541±4.7017 | 1.3668±0.0371 | 1.3345±0.0011 | 0.3292±0.0115 | 0.6516±0.0102 | 63.3995±1.2366 | 1.1954±0.0196 | 1.3291±0.0017 | 0.5845±0.0132 | 1.0000±0.0000 |

| | Stack Overflow | | | | | Yelp | | | | |
|---|---|---|---|---|---|---|---|---|---|---|
| Methods | MAPE(↓) | CRPS(↓) | QQP_Dev(↓) | Top1_ACC(↑) | Top3_ACC(↑) | MAPE(↓) | CRPS(↓) | QQP_Dev(↓) | Top1_ACC(↑) | Top3_ACC(↑) |
| Deter | 4.7518±0.0658 | - | - | 0.5302±0.0010 | 0.8327±0.0014 | 15.9814±2.4486 | - | - | 1.0000±0.0000 | - |
| RMTPP | 7.6946±1.3470 | 6.2844±0.3374 | 1.9317±0.0020 | 0.5343±0.0013 | **0.8555**±0.0073 | 13.6576±0.1261 | 0.0657±0.0005 | 1.3142±0.0055 | 1.0000±0.0000 | - |
| LogNorm | 13.3008±1.2214 | 6.3377±0.2380 | 1.9313±0.0017 | 0.5335±0.0019 | 0.8542±0.0064 | 32.1609±0.3978 | 0.0646±0.0018 | 1.2840±0.0395 | 1.0000±0.0000 | - |
| ERTPP | 17.3008±1.5724 | 4.5747±0.0947 | 1.9299±0.0016 | 0.5316±0.0028 | 0.8526±0.0044 | 34.8405±0.0000 | 0.0673±0.0014 | 1.2632±0.0087 | 1.0000±0.0000 | - |
| WeibMix | 7.6260±1.0663 | 4.3028±0.5535 | 1.9776±0.0000 | 0.5327±0.0011 | 0.8454±0.0056 | 34.8391±0.0019 | 0.0680±0.0000 | 1.2579±0.0390 | 1.0000±0.0000 | - |
| FNNInt | 6.1583±0.0952 | - | 1.5725±0.0065 | 0.5336±0.0009 | 0.8432±0.0010 | 16.2753±0.5204 | - | 1.2579±0.0390 | 1.0000±0.0000 | - |
| SAHP | 5.5246±0.0271 | - | **1.5175**±0.0010 | 0.5235±0.0002 | 0.8278±0.0003 | 12.9830±0.0474 | - | **1.0755**±0.0004 | 1.0000±0.0000 | - |
| THP | 5.6331±0.0413 | - | **1.5033**±0.0016 | 0.5310±0.0003 | 0.8508±0.0001 | 14.4189±0.0474 | - | **1.0775**±0.0005 | 1.0000±0.0000 | - |
| TCDDM | **4.9947**±0.0366 | **0.4375**±0.0163 | 1.5711±0.0043 | **0.5371**±0.0004 | **0.8693**±0.0010 | **10.8426**±0.0253 | **0.0570**±0.0001 | 1.1728±0.0082 | 1.0000±0.0000 | - |
| TCVAE | **5.1397**±0.0626 | **0.5129**±0.0082 | 1.5320±0.0057 | **0.5398**±0.0022 | 0.8418±0.0112 | **9.9204**±0.2895 | 0.0631±0.0008 | **1.1732**±0.0001 | 1.0000±0.0000 | - |
| TCGAN | **5.0874**±0.1527 | 0.5458±0.0254 | **1.5178**±0.0095 | 0.5340±0.0048 | 0.8481±0.0200 | **12.0471**±0.7363 | **0.0608**±0.0022 | 1.1170±0.0275 | 1.0000±0.0000 | - |
| TCCNF | 6.3022±0.0281 | **0.4259**±0.0005 | 1.6319±0.0007 | **0.5428**±0.0003 | **0.8721**±0.0003 | 13.4562±0.2129 | **0.0575**±0.0008 | 1.2355±0.0034 | 1.0000±0.0000 | - |
| TCNSN | 29.4333±2.4937 | 0.8350±0.0035 | 1.6611±0.0004 | 0.5352±0.0012 | 0.8538±0.0095 | 43.9613±2.1338 | 0.4274±0.0131 | 1.5855±0.0055 | 1.0000±0.0000 | - |

The methods of *(1)* ∼ *(4)* have closed-form expectation. Mean of FNNInt, SAHP and THP is obtained by numerical integration, and mean of GNTPP is obtained by Monte Carlo integration thanks to its advantages in flexible sampling. Note that it is possible to sample events from SAHP and THP using Ogata's thinning method (Ogata, 1981) since the intensity for both methods is monotonically decreasing between events. Samples can also be drawn from FNNInt model using numerical root-finding (Shchur et al., 2020a), but these sampling methods designed especially for the three models have not yet been developed. Therefore, flexible sampling is not allowed for FNNInt, SAHP and THP (Table 1), so we do not report their 'CRPS'. In all the generative methods, the trick of **log-noramlization** is used: The input samples are firstly normalized by $\frac{\log \tau - \text{Mean}(\log \tau)}{\text{Var}(\log \tau)}$ during training, and rescaled back by $\exp(\log \tau \text{Var}(\log \tau) + \text{Mean}(\log \tau))$ to make sure the sampled time intervals are positive.

For fair comparison, we all use revised attentive encoder which is a variant of Transformer to obtain the historical encodings.

We conclude from the experimental results in Table 3 that

- All these established generative methods show comparable effectiveness and feasibility in TPPs, except TCNSN as a score matching method. TCDDM, TCVAE and TCGAN usually show good performance in next arrival time prediction, compared with the diffusion decoder.

- In spite of good performance, as a continuous model, TCCNF is extremely time-consuming, whose time complexity is unaffordable as shown in Appendix B.4.

- By using the numerical integration to obtain the estimated expectation of SAHP and THP, we find they can also reach comparable 'MAPE' to generative decoders. However, the two models do not provide a flexible sampling methods, where the time interval samples cannot be flexibly drawn from the learned conditional distribution.

- From 'CRPS' and 'QQP_Dev' evaluating models' fitting abilities of arrival time, the generative decoders still outperforms others. For 'QQP_Dev', SAHP and THP's show very competitive fitting ability thanks to its employing expressive formulation as the intensity function.

For the `Synthetic` dataset, results are given in Appendix B.3. In summary, the empirical results show that proposed generative neural temporal point process employs and demonstrates deep generative models' power in modeling the temporal point process.

**Further Discussion on Generative Models.** As TCDDM, TCCNF, and TCNSN can all be classified into score-based methods according to Song et al. (2021b), in which they are described as different stochastic differential equations, this raises a question to us: Why only TCNSN fails to model the temporal point process effectively? Following the former work (Song et al., 2021b; Song & Ermon, 2019; 2020), the continuous form of temproal conditional score-matching model is given by the stochastic differential equation (SDE) as

$$dτ = \sqrt{\frac{d[σ^2(k)]}{dk}}dw, \tag{30}$$

where $w$ is a standard Wiener process. It is a variance exploding process because the variance goes to infinity as $k \to +\infty$. In comparison, the forward process of temporal conditional diffusion model can be regarded as a variance perserving one, as

$$dτ = -\frac{1}{2}\sqrt{1-β(k)}τdk + \sqrt{β(k)}dw. \tag{31}$$

And temporal conditional continuous normalizing flow is the deterministic process where $dw = 0$, as

$$dτ = f_θ(τ, k)dk, \tag{32}$$

where $f_θ$ is a learnable neural network. Note that we omit the conditional notation in the single event modeling for simplicity.

In the reverse process which is used for sampling (or denoising), these three models firstly sample $τ_K \sim \mathcal{N}(0, 1)$. $τ_K$ is denoised by the learnable score function $ε_θ$ in TCDDM and TCNSN, or invertible process in TCCNF, and $τ_0$ as the output of the final stage of the process is generated as the time interval sample. For the variance exploding property of TCNSN, in the reverse process as shown in Figure 5, the variance of the distribution will firstly become excessively large. As a result, later in small-variance scales, it cannot recover the input signals and distributions attributing to the high variance of the early stage. In comparison, the variance in the sampling dynamics of TCDDM and TCCNF keeps stable, and the learned distributions are approaching the input gradually in the iteration of denoising process.

## 5.3 Advantages of Revised Encoders

In this part, we aim to conduct empirical study to prove the better expressivity our revised attentive encoder. We first fix the probabilistic decoder as diffusion decoder, and conduct experiments with different history encoders, including our revised attentive (REV-ATT), self-attentive (ATT) and LSTM encoders to demonstrate the advantages in expressiveness of the revised attention. Table 4 shows the advantages of the

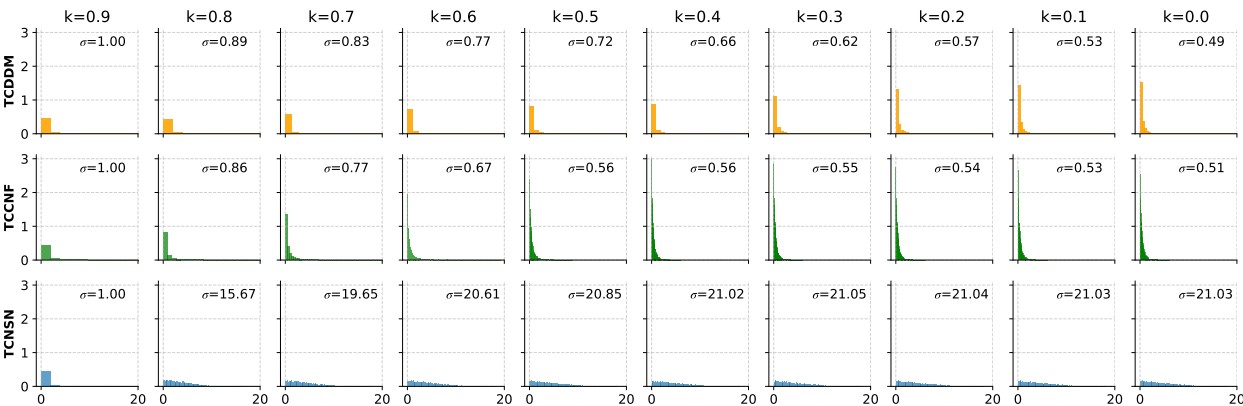

Figure 5: The empirical distribution of the sample generating dynamics of TCDDM, TCCNF, and TCNSN. The visualization is conducted in `Stack Overflow` dataset obtained by 5000 samples, with the iteration step set as 100 in TCDDM and 1000 in TCNSN. We regard the time scale as $[0, 0.9]$, and give the dynamics of the distribution change of the reverse sampling process at different discrete time points. As demonstrated, the variance of TCNSN is much larger than others, which prevents the model from recovering the distribution of the input samples. **Log-noramlization** trick is used to force the intermediate samples to be positive while $\sigma$ is calculated with unnormalized latent variables for consistent order of numerical values.

Table 4: Comparison of different history encoders.

| | | MOOC | | | |
|---|---|---|---|---|---|
| Encoders | MAPE | CRPS | QQP_Dev | Top1_ACC | Top3_ACC |
| LSTM | $23.3562_{\pm0.0076}$ | $0.1468_{\pm0.0000}$ | $1.0369_{\pm0.0000}$ | $0.4232_{\pm0.0004}$ | $0.7279_{\pm0.0001}$ |
| ATT | $\mathbf{23.3559}_{\pm0.0283}$ | $0.1468_{\pm0.0000}$ | $1.0369_{\pm0.0000}$ | $0.4228_{\pm0.0003}$ | $0.7275_{\pm0.0000}$ |
| Rev-Att | $\mathbf{23.3559}_{\pm0.3098}$ | $0.1468_{\pm0.0000}$ | $1.0369_{\pm0.0000}$ | $\mathbf{0.4308}_{\pm0.0069}$ | $\mathbf{0.7408}_{\pm0.0044}$ |

| | | Retweet | | | |
|---|---|---|---|---|---|
| Encoders | MAPE($\downarrow$) | CRPS($\downarrow$) | QQP_Dev($\downarrow$) | Top1_ACC($\uparrow$) | Top3_ACC($\uparrow$) |
| LSTM | $16.3525_{\pm0.0237}$ | $0.2079_{\pm0.0001}$ | $1.0521_{\pm0.0012}$ | $0.6083_{\pm0.0002}$ | $1.0000_{\pm0.0000}$ |
| ATT | $16.3160_{\pm0.0397}$ | $0.2077_{\pm0.0001}$ | $1.0469_{\pm0.0002}$ | $0.6083_{\pm0.0001}$ | $1.0000_{\pm0.0000}$ |
| Rev-Att | $\mathbf{15.6058}_{\pm0.5550}$ | $\mathbf{0.2076}_{\pm0.0001}$ | $\mathbf{1.0327}_{\pm0.0111}$ | $\mathbf{0.6274}_{\pm0.0075}$ | $1.0000_{\pm0.0000}$ |

| | | Stack Overflow | | | |
|---|---|---|---|---|---|
| Encoders | MAPE($\downarrow$) | CRPS($\downarrow$) | QQP_Dev($\downarrow$) | Top1_ACC($\uparrow$) | Top3_ACC($\uparrow$) |
| LSTM | $5.0381_{\pm0.0055}$ | $0.4502_{\pm0.0005}$ | $1.5737_{\pm0.0006}$ | $0.5337_{\pm0.0001}$ | $0.8626_{\pm0.0001}$ |
| ATT | $5.0285_{\pm0.0290}$ | $0.4502_{\pm0.0012}$ | $\mathbf{1.5683}_{\pm0.0013}$ | $0.5326_{\pm0.0002}$ | $0.8632_{\pm0.0001}$ |
| Rev-Att | $\mathbf{4.9947}_{\pm0.0366}$ | $\mathbf{0.4375}_{\pm0.0163}$ | $1.5711_{\pm0.0043}$ | $\mathbf{0.5371}_{\pm0.0004}$ | $\mathbf{0.8693}_{\pm0.0010}$ |

| | | Yelp | | | |
|---|---|---|---|---|---|
| Encoders | MAPE | CRPS | QQP_Dev | Top1_ACC | Top3_ACC |
| LSTM | $10.9082_{\pm0.0387}$ | $0.0571_{\pm0.0001}$ | $1.1792_{\pm0.0003}$ | $1.0000_{\pm0.0000}$ | - |
| ATT | $10.9119_{\pm0.0188}$ | $0.0571_{\pm0.0001}$ | $1.1792_{\pm0.0002}$ | $1.0000_{\pm0.0000}$ | - |
| Rev-Att | $\mathbf{10.8426}_{\pm0.0253}$ | $\mathbf{0.0570}_{\pm0.0001}$ | $\mathbf{1.1728}_{\pm0.0082}$ | $1.0000_{\pm0.0000}$ | - |

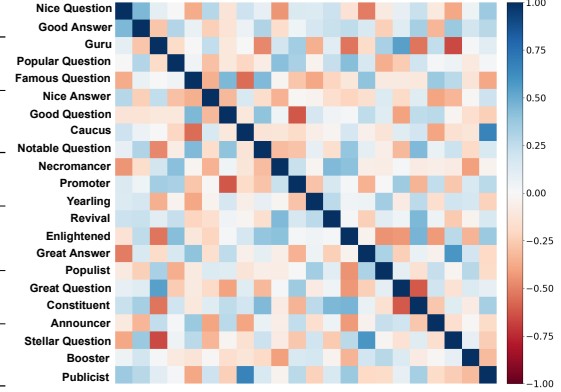

Figure 6: The relations of similarity between event types of `Stack Overflow` which is inferenced by Rev-Att + TCDDM. Rows are arranged in the same order as columns.

revision on two datasets, the revised attention outperforms others in most metrics. Results on `Synthetic` dataset are shown in Appendix B.3.

The revised attentive encoder achieves overall improvements by a small margin. The complete empirical study (Lin et al., 2021) has illustrate that the performance gain brought from history encoders is very small, and our revision can further brings tiny improvements.

Second, we give visualization shown in Figure 6 on the events' relations of similarity obtained by $\boldsymbol{E}_m \boldsymbol{E}_m^T$, as discussed in Section 3.1. It shows that the effects of some pairs of event types are relatively significant with high absolute value of event similarity, such as (*Stellar Question*, *Great Answer*) and (*Great Question*, *Constituent*). It indicates the statistical co-occurrence of the pairs of the event types in a sequence.

### 5.4 Hyperparameter Sensitivity Analysis

Several hyper-parameters affect the model performance, and in this part we try to explore their impacts. We conduct experiments on different '*embedding size*', i.e. *D* and '*layer number*'. The partial results of TCDDM are given in Figure 7 and 8, and details are shown in Appendix B.4.

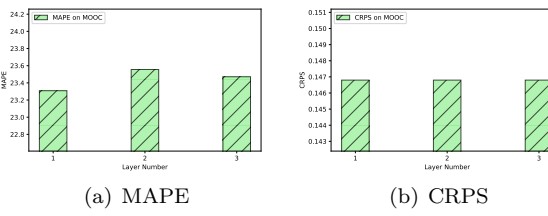

(a) MAPE          (b) CRPS

Figure 7: Change of Performance with *layer number* of TCDDM on `MOOC`.

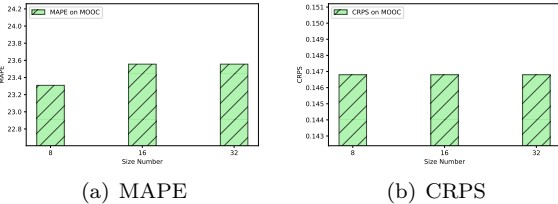

(a) MAPE          (b) CRPS

Figure 8: Change of Performance with *embedding size* of TCDDM on `MOOC`.

The MAPE metric is more sensitive than CRPS with the change of the hyperparameter of the model. In `MOOC` dataset, the large 'layer number' and 'embedding size' is not beneficial to predictive performance.

## 6 Conclusion

A series of deep neural temporal point process (TPP) models called GNTPP, integrating deep generative models into the neural temporal point process and revising the attentive mechanisms to encode the history observation. GNTPP improves the predictive performance of TPPs, and demonstrates its good fitting ability. Besides, the feasibility and effectiveness of GNTPP have been proved by empirical studies. And experimental results show good expressiveness of our revised attentive encoders, with events' relation provided.

A complete framework with a series of methods are integrated into our code framework, and we hope the fair empirical study and easy-to-use code framework can make contributions to advancing research progress in deep neural temporal point process in the future.

## Acknowledgement

This work is supported in part by National Natural Science Foundation of China, Geometric Deep Learning and Applications in Proteomics-Based Cancer Diagnosis (No. U21A20427). We thank a lot to all the reviewers who are responsible, careful and professional in TMLR for their valuable and constructive comments.

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

# A    Preliminaries on Temporal Point Process

**Temporal point process with markers.** For a temporal point process $\{t_i\}_{i\geq 1}$ as a real-valued stochastic process indexed on $\mathbb{N}^+$ such that $T_i \leq T_{i+1}$ almost surely (here $T_i$ representing the random variable), each random variable is generally viewed as the arrival timestamp of an event. When each timestamp is given a type marker, i.e. $\{(t_i, m_i)\}_{i\geq 1}$, the process is called marked temporal point process, also called multivariate

point process as well.

**Conditional intensity function and probability density function.** As defined in Eq. 1, the probability density function and cumulative distribution function can be obtained through

$$
\begin{aligned}
\lambda(t|\mathcal{H}(t))dt &= \mathbb{E}[\mathcal{N}(t+dt) - \mathcal{N}(t)|\mathcal{H}(t)] \\
&= \mathbb{P}(t_i \in [t, t+dt)|\mathcal{H}(t)) \\
&= \mathbb{P}(t_i \in [t, t+dt)|t_i \notin [t_{i-1}, t), \mathcal{H}(t_{i-1})) \\
&= \frac{\mathbb{P}(t_i \in [t, t+dt), t_i \notin [t_{i-1}, t)|\mathcal{H}(t_{i-1}))}{\mathbb{P}(t_i \notin [t_{i-1}, t)|\mathcal{H}(t_{i-1})))} \\
&= \frac{\mathbb{P}(t_i \in [t, t+dt)|\mathcal{H}(t_{i-1}))}{\mathbb{P}(t_i \notin [t_{i-1}, t)|\mathcal{H}(t_{i-1}))} \\
&= \frac{p(t|\mathcal{H}(t_{i-1}))}{1 - P(t|\mathcal{H}(t_{i-1}))} \\
&= \frac{p^*(t)}{1 - P^*(t)},
\end{aligned}
$$

In this way, the reverse relation can be given by

$$
p^*(t) = \lambda^*(t) \exp(-\int_{t_{i-1}}^{t} \lambda^*(\tau)d\tau);
$$

$$
P^*(t) = 1 - \exp(-\int_{t_{i-1}}^{t} \lambda^*(\tau)d\tau).
$$

**Example 1.** (Poisson process) The (homogeneous) Poisson process is quite simply the point process where the conditional intensity function is independent of the past. For example, $\lambda^*(t) = \lambda(t) = c$ which is a constant.

**Example 2.** (Hawkes process) The conditional intensity function of which can be written as

$$
\lambda^*(t) = \alpha + \sum_{t_j < t} g(t - t_j; \eta_j, \beta_j),
$$

which measures all the impacts of all the historical events on the target timestamp $t$. The classical Hawkes process formulates the impact function $g(t - t_j; \eta, \beta) = \eta \exp(\beta(t - t_j))$ as the exponential function.

## B    Experiments

### B.1    Sythetic Dataset Description

The synthetic datasets are generated by `tick`[1] packages (Bacry et al., 2017), using the Hawkes process generator. Four Hawkes kernels as impact functions are used with each process's intensity simulated according to **Example 2**, including

$$
\begin{aligned}
g_a(t) &= 0.09 \exp(-0.4t) \\
g_b(t) &= 0.01 \exp(-0.8t) + 0.03 \exp(-0.6t) + 0.05 \exp(-0.4t) \\
g_c(t) &= 0.25|\cos 3t| \exp(-0.1t) \\
g_d(t) &= 0.1(0.5 + t)^{-2}
\end{aligned}
$$

The impact function $g_{j,i}(t)$ measuring impacts of type $i$ on type $j$ is uniformly-randomly chosen from above. A probability equalling to $r$ which we called *cutting ratio* is set to force the impact to zero, thus leading the Granger causality graph to be sparse. The cutting ratio is set as 0.2, and the total number of types is set as 5.

---

[1]https://github.com/X-DataInitiative/tick

## B.2 Calculation of QQP_Dev

If the sequences come from the intensity function of point process $\lambda(t)$ , then the integral $\Lambda = \int_{t_i}^{t_{t+1}} \lambda(\tau) d\tau$ between consecutive events should be exponential distribution with parameter 1. Therefore, the QQ plot of $\Lambda$ against exponential distribution with rate 1 should fall approximately along a 45-degree reference line. Therefore, we first use the model to sample a series timestamps, and use them to estimate the empirical $\Lambda^*$. Mean absolute deviation of the QQ plot of it v.s $\text{Exp}(1)$ from the line with slop 1 is 'QQP_DEV'.

## B.3 Supplementary Results

We first give supplementary results of different methods on the `Sythetic` dataset shown by Table 5.

Table 5: Comparison on `Sythetic` dataset.

| Methods | Synthetic | | | | |
| --- | --- | --- | --- | --- | --- |
| | MAPE | CRPS | QQP_Dev | Top1_ACC | Top3_ACC |
| E-RMTPP | $22.8206_{\pm1.5594}$ | $0.1905_{\pm0.0110}$ | $1.7921_{\pm0.0047}$ | $0.2497_{\pm0.0031}$ | $0.6693_{\pm0.0034}$ |
| LogNorm | $54.6208_{\pm0.0000}$ | $0.1916_{\pm0.0102}$ | $1.7920_{\pm0.0044}$ | $0.2494_{\pm0.0027}$ | $0.6687_{\pm0.0026}$ |
| E-ERTPP | $54.6208_{\pm0.0000}$ | $0.1843_{\pm0.0072}$ | $1.7881_{\pm0.0062}$ | $0.2482_{\pm0.0011}$ | $0.6674_{\pm0.0012}$ |
| WeibMix | $26.5910_{\pm7.3426}$ | $0.1060_{\pm0.0029}$ | $1.9776_{\pm0.0000}$ | $0.2476_{\pm0.0001}$ | $0.6668_{\pm0.0002}$ |
| FNNInt | $4.5223_{\pm0.0976}$ | - | $1.3342_{\pm0.0005}$ | $0.2531_{\pm0.0041}$ | $0.6724_{\pm0.0040}$ |
| SAHP | $4.5198_{\pm0.1677}$ | - | $1.1775_{\pm0.0002}$ | $0.2964_{\pm0.0003}$ | $0.7269_{\pm0.0001}$ |
| THP | $4.4958_{\pm0.1331}$ | - | $1.1775_{\pm0.0001}$ | $0.2474_{\pm0.0002}$ | $0.6667_{\pm0.0002}$ |
| TCVAE | $3.3237_{\pm0.0304}$ | $0.0617_{\pm0.0001}$ | $1.4124_{\pm0.0024}$ | $0.2476_{\pm0.0002}$ | $0.6670_{\pm0.0000}$ |
| TCGAN | $3.5009_{\pm0.1288}$ | $0.2510_{\pm0.2690}$ | $1.4297_{\pm0.0223}$ | $0.1924_{\pm0.0894}$ | $0.5059_{\pm0.2438}$ |
| TCCNF | $4.7095_{\pm0.0303}$ | $0.0654_{\pm0.0000}$ | $1.5787_{\pm0.0007}$ | $0.2465_{\pm0.0001}$ | $0.6669_{\pm0.0001}$ |
| TCNSN | $33.9541_{\pm0.9428}$ | $0.1109_{\pm0.0002}$ | $1.5884_{\pm0.0001}$ | $0.2554_{\pm0.0001}$ | $0.6766_{\pm0.0001}$ |
| TCDDM | $3.2323_{\pm0.0015}$ | $0.0509_{\pm0.0000}$ | $1.4261_{\pm0.0002}$ | $0.2492_{\pm0.0020}$ | $0.6686_{\pm0.0019}$ |

We give the negative ELBO which is the upper bound of models' NLL of TCDDM, TCVAE, and TCNSN, and the exact NLL of other models except that in TCGAN we give Wasserstein distance between the empirical and model distributions in the four real-world dataset, as shown in Table 6.

## B.4 Complexity Comparison

We provide each methods mean training time for one epoch to figure out which methods are extremely time-consuming. It shows all these methods are affordable in time complexity except `CNSN`, and `CGAN` is also time-consuming. The test is implemented on a single Nvidia-V100(32510MB). In all the test setting, batch size is set as 16, embedding size is 32 and layer number is 1. For methods whose likelihood has no closed-form, we use Monte Carlo integration, where in each interval, the sample number is 100.

Table 6: Comparison on four real-world datasets on the NLL and ELBO.

| Methods | MOOC | Retweet | Stack Overflow | Yelp |
| --- | --- | --- | --- | --- |
| E-RMTPP | $1.7504$ | $-1.9872$ | $4.9031$ | $-1.0832$ |
| LogNorm | $1.3635$ | $-2.4197$ | $4.8782$ | $-1.2808$ |
| E-ERTPP | $3.5791$ | $-0.8876$ | $5.0845$ | $-0.9678$ |
| WeibMix | $0.7950$ | $-2.5110$ | $3.8717$ | $-1.1125$ |
| FNNInt | $-2.3024$ | $-3.0064$ | $1.8469$ | $-1.2294$ |
| SAHP | $-2.3472$ | $-2.9955$ | $1.8348$ | $-1.6607$ |
| THP | $0.1270$ | $-1.3794$ | $1.8591$ | $-1.6349$ |
| TCDDM | $\leq 1.7609$ | $\leq 0.7560$ | $\leq 2.2450$ | $\leq 0.0142$ |
| TCVAE | $\leq 9.4754$ | $\leq 7.5911$ | $\leq 3.9727$ | $\leq 6.2181$ |
| TCGAN | $(0.0056)$ | $(0.0001)$ | $(0.0511)$ | $(0.0560)$ |
| TCCNF | $-2.8591$ | $-3.0464$ | $1.6881$ | $-1.8791$ |
| TCNSN | $\leq 2.1815$ | $\leq 0.8340$ | $\leq 2.4131$ | $\leq 0.3517$ |

Table 7: Comparison of time complexity.

| Methods | Time per Epoch | | | | |
|---|---|---|---|---|---|
| | MOOC | Retweet | Stack Overflow | Yelp | Synthetic |
| E-RMTPP | 46″ | 2′08″ | 1′22″ | 12″ | 1′26″ |
| LogNorm | 39″ | 2′02″ | 1′14″ | 11″ | 1′27″ |
| E-ERTPP | 42″ | 2′14″ | 1′17″ | 11″ | 1′33″ |
| WeibMix | 49″ | 2′18″ | 1′27″ | 13″ | 1′22″ |
| FNNInt | 45″ | 2′08″ | 1′16″ | 14″ | 1′32″ |
| SAHP | 1′11″ | 2′42″ | 1′51″ | 21″ | 2′14″ |
| THP | 57″ | 2′29″ | 1′30″ | 18″ | 2′02″ |
| TCVAE | 46″ | 2′43″ | 1′33″ | 11″ | 1′31″ |
| TCGAN | 2′30″ | 11′24″ | 3′47″ | OOM | 4′02″ |
| TCCNF | 5′42″ | 21′28″ | 7′06″ | 3′24″ | 5′47″ |
| TCNSN | 33″ | 1′46″ | 42″ | 10″ | 56″ |
| TCDDM | 35″ | 1′39″ | 1′16″ | 13″ | 1′12″ |

Table 8: Comparison of used memory.

| Methods | Peak Memory in Training | | | | |
|---|---|---|---|---|---|
| | MOOC | Retweet | Stack Overflow | Yelp | Synthetic |
| E-RMTPP | 2174 MiB | 1544 MiB | 4288 MiB | 22294 MiB | 7074 MiB |
| LogNorm | 1976 MiB | 1544 MiB | 4096 MiB | 22294 MiB | 7078 MiB |
| E-ERTPP | 1976 MiB | 1544 MiB | 4096 MiB | 22294 MiB | 7078 MiB |
| WeibMix | 2078 MiB | 1544 MiB | 4290 MiB | 22294 MiB | 7080 MiB |
| FNNInt | 1828 MiB | 1438 MiB | 3692 MiB | 21366 MiB | 10686 MiB |
| SAHP | 7197 MiB | 1606 MiB | 4706 MiB | 22730 MiB | 15258 MiB |
| THP | 6004 MiB | 1496 MiB | 4544 MiB | 18384 MiB | 7648 MiB |
| TCVAE | 2286 MiB | 1436 MiB | 3198 MiB | 27684 MiB | 6854 MiB |
| TCGAN | 3712 MiB | 2164 MiB | 4906 MiB | OOM | 12318 MiB |
| TCCNF | 2598 MiB | 1486 MiB | 3932 MiB | 27836 MiB | 4048 MiB |
| TCNSN | 2926 MiB | 1662 MiB | 3884 MiB | 22374 MiB | 7984 MiB |
| TCDDM | 3110 MiB | 1542 MiB | 3508 MiB | 22764 MiB | 8084 MiB |

## B.5 Hyperparameter Sensitivity Analysis

We give the CDDM's performance under different parameters on MOOC, Retweet and Stack Overflow, with layer number and embedding size set in range of $\{1, 2, 3\}$ and $\{8, 16, 32\}$ respectively. It shows that the large embedding size usually brings improvements, so we recommend that it should be set as 32. And layer number should be set as 1 or 2.

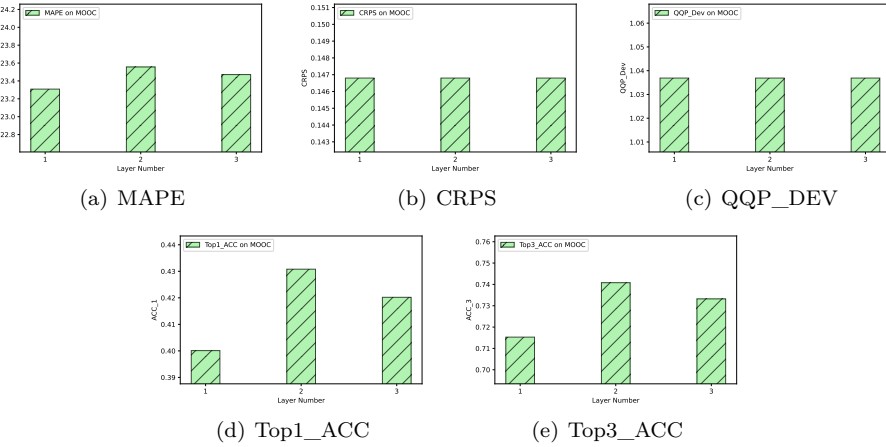

(a) MAPE      (b) CRPS      (c) QQP_DEV

(d) Top1_ACC      (e) Top3_ACC

Figure 9: Change of Performance with Layer Number on MOOC

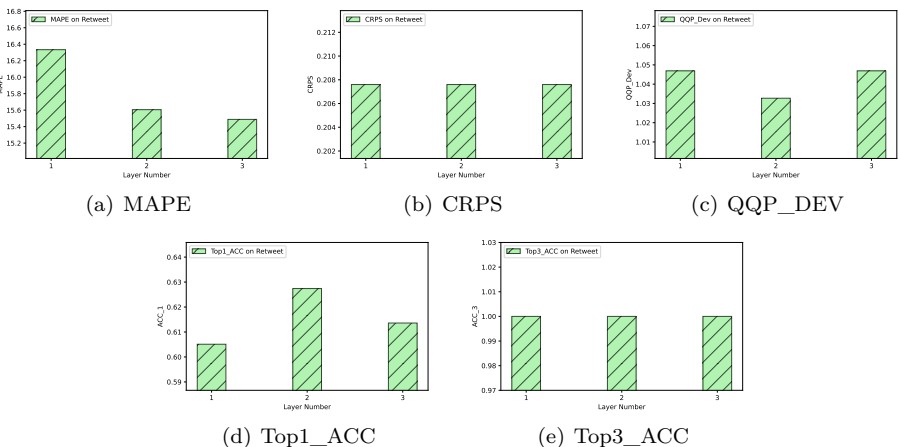

Figure 10: Change of Performance with Layer Number on `Retweet`

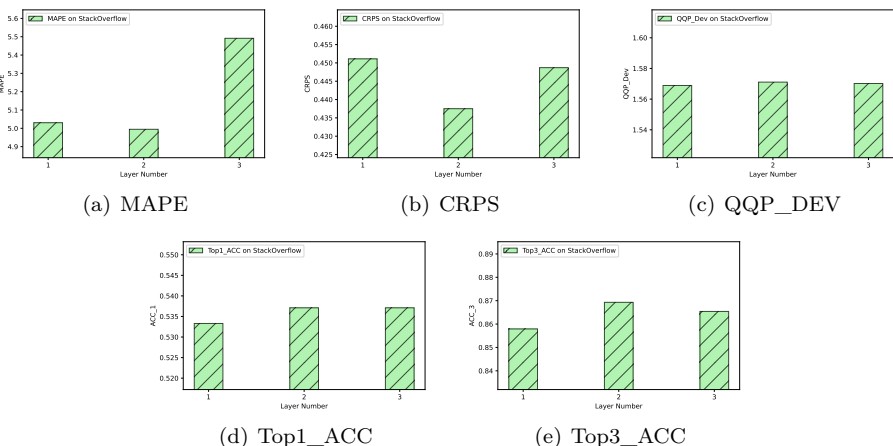

Figure 11: Change of Performance with Layer Number on `Stack Overflow`

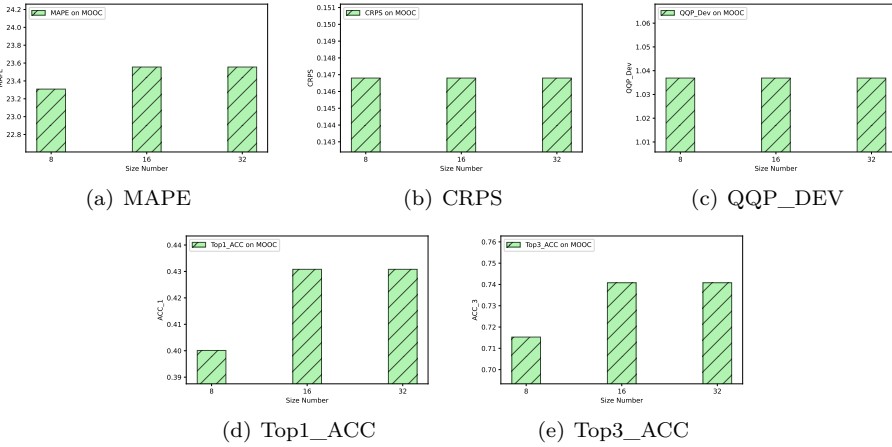

Figure 12: Change of Performance with embedding size on `MOOC`

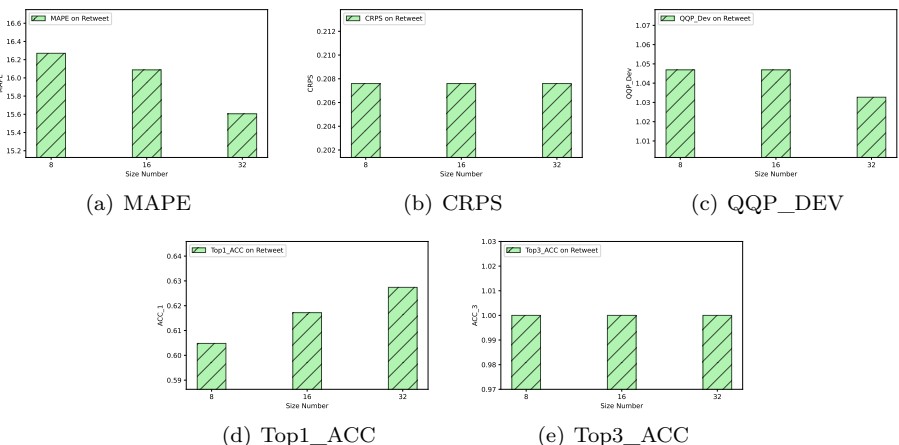

Figure 13: Change of Performance with embedding size on `Retweet`

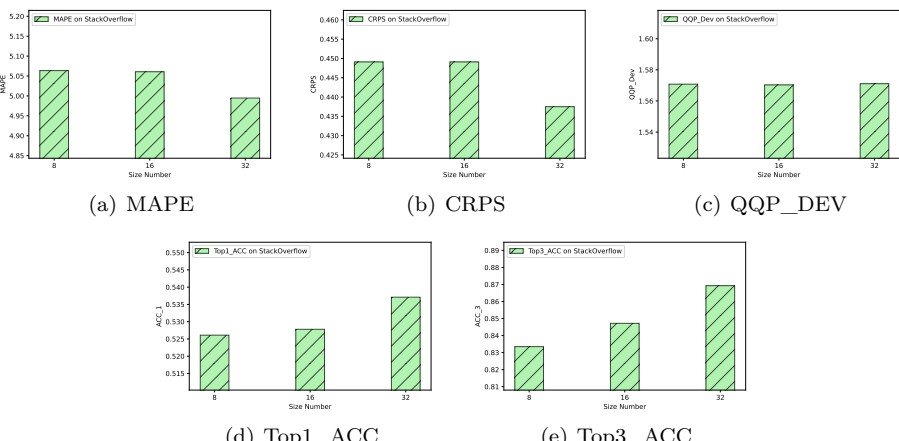

Figure 14: Change of Performance with embedding size on `StackOverflow`

