# OpenReview forum: "Exploring Generative Neural Temporal Point Process"
_TMLR — Accepted by TMLR_

### Review · Reviewer_Y3mv · 2022-06-04

**Summary Of Contributions:**

**Summary of the contributions:** The authors propose a generative framework for neural temporal point process (GNTPP) to apply deep generative models in the context of Temporal Point Processes. The paper breaks down previous research into Temporal Point processes into two parts - historical encoders to generate context from observations and probabilistic decoders that generate a distribution over the next event time conditioned on this historical context. The authors then propose a modification to the attention mechanism of the historical encoder proposed by Zhang et. al and Zuo et. al to incorporate additional context from event types and study different conditional generative models (Diffusion/VAE/WGAN/ Score based/Flows) to learn the distribution of next event time conditioned on this historical context vector.


**Broader Impact Concerns:**

**Broader impact concerns:** None

**Requested Changes:**

**Requested changes:** Weakness 1, 2, 3, 4 need to be addressed by the authors and are critical for securing my recommendation. Weakness 5 and other concerns 5 would make the paper stronger.



**Strengths And Weaknesses:**

**Strengths:** Application of generative models to Temporal Point processes is an interesting line of research and it is interesting to study if they provide a viable alternative to traditional conditional intensity based modelling. The authors study various families of generative models and the comparison would be appreciated by the community.

**Weakness:** The authors make certain claims that are not backed up by evidence and certain claims that are false. The following are the claims that I have issues with:

1. In the abstract and introduction, The authors claim previous works have focused on maximizing the likelihood of the next event which leads to poor predictive performance and that generative models can be used to improve performance. Again in section 2.1.3 the authors claim *“Differing from the previous works focusing on models’ fitting ability in terms of higher likelihood, these models aim to promote models’ prediction ability, i.e. to generate high-quality samples which are closer to ground truth observations”*.
I do not quite understand what the authors are trying to imply. All the generative models that the authors have proposed either maximize a bound on the log-likelihood of the predicted event which surmounts to minimizing the KL divergence between the true and the predicted distribution (VAE/Diffusion/Score) or the Wasserstein distance (WGAN). The authors also do not convey why this is expected to improve the predictive performance. This brings me to my second point.

2. If improving predictive performance is something the authors are interested in, an important baseline to be considered is adding this as an explicit penalty term along with the Negative log-likelihood loss as done in Transformers Hawkes Process (THP) (Zuo et. al).

3. In section 2.2, the authors claim the Monte Carlo estimate used in THP and SAHP is biased. This is false, as the THP authors mention that the estimate is unbiased and I have verified it to the best of my knowledge.

4. In section 2.3, to motivate the revised encoder, P1 mentions use time difference instead of actual event time to show a scenario where the embeddings are the same. This seems wrong as $j_1$ and $j_2$ are different, $\omega(\tau_{j_1})$ and $\omega(\tau_{j_2})$ are going to be different even if $\tau_{j_1}$ and $\tau_{j_2}$ are the same.

5. In section 2.1.3, the authors make a comment about the computational cost of the monte carlo integral used in THP and SAHP. As the authors claim their method avoids this cost, It would be interesting to look at a comparison of FLOPs/ Wall-clock time for training/ inference as generative models come with their own overhead.

6. The authors do not mention whether the baseline RMTPP uses the same time normalization they mention in section 4.1.2. Any scaling of the time of the events leads to a change in the negative log-likelihood by a constant factor and makes the comparison meaningless.

**Other Concerns:**
1. In section 2.3 the authors talk about transformers yet do not cite the original paper.

2. The citation for Transformers Hawkes process which is heavily featured in the paper is incomplete.

3. In section 2.6, The authors should mention their assumption that the mark and time distributions are conditionally independent given the historical embedding

4. In section 4.2, the authors use an example from the tick library as their simulated example thus I think it is important to cite the work in the main paper.

5. It would be nice to see the actual QQPlot to see how close it is to the $y=x$ line instead of just the deviation as done in SAHP.

6. Section 4.2, line 1 is missing the word “decoder”.

---

> ### Author Response · Authors · 2022-06-06
> **Responses and Paper of New Version**
>
> Thanks for your valuable comments. According to them, we revised our paper, which has been uploaded.  Here are our responses, and you can read them together with the new version.
>
> 1. First, in existing works, the TPP models as a branch of probabilistic models, aim to obtain a lower NLL to fit the data better, and the NLL is usually regarded as the most important metric to judge the performance. However, in the generative task, the samples generated by these models are very far from the ground truth observations, which has been recently shown by a complete fair empirical study (Lin et al. 2021). Therefore, we think that a good TPP model should not only demonstrate its goodness-of-fitting (lower NLL) but also be able to generate arrival time as samples of high quality.
> Motivated by this, we think we can borrow the SOTA techniques in generative models to improve the quality of samples: (1) enable the expectation of learned distribution similar to ground truth (better as lower MAPE and first term in CRPS) (2) preserve the randomness and diversity of the generated samples (better as higher second term in CRPS). That’s why we choose the two metrics other than NLL. We add more description in line.51-53, and hope it may not confuse you and other readers a lot.
> To differentiate the predictive performance in our paper and others in the works like Transformer, in our paper it is defined in the setting of probabilistic density estimation, as to generate high-quality samples which are closer to ground truth observations. The probabilistic decoders generate a series of values $(\hat t_k)_{k\leq N}$, and then use the mean as the expectation $\mathbb{E}(t)$ for prediction, while $Var(t) \neq 0$. Unlike Transformer, neural TPP is still a generative task as density estimation instead of a regression task as point estimation.  While in THP (Zuo et al., 2021), there is another head to output $\hat t$ as the arrival-time prediction, and the loss for this head is set as a summation of difference, which is a deterministic process, as $\hat t_k$ are all the same.
>
> 2. The penalty term as extra loss in THP is summation of difference  (see line. 62 in its source code : https://github.com/SimiaoZuo/Transformer-Hawkes-Process/blob/master/Main.py ), and in evaluating, it uses the output of the head as the predicted next-arrival-time (line 113 in the given link). Obviously, this head cannot generate samples of diversity. Therefore, the evaluation of predictive performance in our experiment and THP is different, because ours are in the evaluating settings of probabilistic models and THP is in the setting of deterministic models.
> The baseline we choose in the first version does not include THP and SAHP, because they have no closed likelihood form. We added them in the second version, see line.364 to 366, Table. 3 for details. Note that in these two methods, we still use the setting in probabilistic models, i.e. using the expectation of learned probability distribution as the predicted next-arrival-time, rather than the output value from the deterministic head. Because these two methods do not allow flexible sampling, we use numerical integration. CRPS cannot be obtained because they do not allow flexible sampling.
>
> 3. We are very sorry for our abuse of terms. We want to express that the deviation of the approximation with Monte Carlo integration from the analytical likelihood may occur, rather than accuse that the stochastic integration is biased. We have revised the error (see line. 137 and 138).
>
> 4. Our motivation originates from THP’s time embedding (Eq.2 in THP), where the position term $j$ is not considered. In SAHP, the position term can fix Problem 1, and it is our negligence and carelessness to not differentiate the two methods’ time embedding in our context. In our new version, we revise the context, as shown in line.169, line.178 and line 192. And in implementation, our time-embedding considers both time and position, while the exponential term in attentive encoder can further improve the performance slightly. As the revision of attention mechanism is not our main contribution, we hope this fixed mistake won’t cause a negative impact on your evaluation of our work.
>
> 5. We add a simple comparison of methods in 2.2 including time cost, memory cost, and their unsatisfactory predictive performance in Figure.1, to give an intuitive demonstration of the motivation of our work.
>
> 6. The methods are implemented in a unifying framework, you can download the supplementary to run them, including RMTPP. The input data are all normalized into [0, 50], for numerical stability. (For example, in MAPE, when $t_i – t_{i-1}$ is too small, the metric will be meaningless.) We check it again and definitely assure that the comparison in our experiment is fair.
>
> Other concerns:
> We have added the mentioned citations and the basic assumption in our new version.
> For example, line.285, line 530-533, and line 419.
> Thanks a lot for your valuable advice.

---

> > ### Comment · Reviewer_Y3mv · 2022-06-22
> > **Response to Authors**
> >
> > I thank the authors for the prompt reply and changes to the manuscript. Points 3-6 and other concerns seem to be addressed by the authors and I have no further questions regarding them. However, I still have some more questions.
> >
> > 1. I understand what the authors are trying to imply however I still have the following reservations. The authors claim that *“TPP models should not only demonstrate its goodness of fitting but also be able to generate samples of high quality”.*
> >
> > What I find missing and think the community would appreciate is **Why do we expect these generative models to do well on predictive metrics and why do they perform well on these metrics when their training involves maximizing ELBO  (minimizing NLL)?**
> >
> > High-quality sample generation requires learning the underlying conditional density function $p(\tau | \boldsymbol{h}_i )$ that shows the distribution of event time given the event history. Minimizing NLL is equivalent to minimizing the KL divergence between the learned and the underlying conditional density function which is performed by the current line of works THP, SAHP. The methods authors propose, also surmount to minimizing NLL (or maximizing ELBO) during training, thus bringing forth the question why are the proposed methods better at predictive metrics like MAPE and CRPS. In my opinion (and please do correct me if I am wrong), the improvement seems to arise from the fact that the proposed generative models circumvent assuming a certain analytical form for the conditional intensity function $\lambda^*(t)$ (which is done by THP and SAHP) allowing for modeling a richer class of conditional density functions. I would appreciate some more discussion regarding this.

---

> > > ### Author Response · Authors · 2022-06-23
> > > **Responses**
> > >
> > > 1. Our motivation comes from the MAPE metric which is widely used in THP, SAHP, EDTPP and so on.
> > > We first found that their MAPE are all very high, and as a probabilistic model, the statistical inference ability is definitely very important, as a series of probabilistic time series models demonstrate [1, 2]. Therefore, it is natural to think of the existing methods can hardly generate samples that are close to the ground truth observation. If the likelihood is enough to demonstrate a model's generative performance, I think FID and other metrics in image generation in computer vision are not necessary. However, in computer vision, to judge a model's ability to generate high-quality samples, there are a lot of metrics proposed, and generative models are not limited to be compared in only NLL. Although MAPE is naive, but it can reflect the generated sample's quality. And CRPS in time series modelling is used more widely, which can also reflect sample quality and fitting ability.
> > >
> > > 2. I agree with you that the generative models outperform THP and SAHP because they allow for modeling a richer class of conditional density functions, which is also mentioned in our introduction parts. And as we discussed, this paper is to `EXPLORING' their feasibility rather than replace all the previous methods due to their state-of-the-art performance. We admit that the novelty is not astonishing, with some revision on existing generative methods and the improvements are not significant. However, the empirical study is complete and the code framework is easy to use with discussed methods all integrated, which we think meet the tenet of TMLR.
> > > Besides, THP and SAHP donot allow flexible generation, which is also a weakness of these models. For application scenarios that require the model to generate next-event-time, they are limited.
> > >
> > > 3. More theoretical and empirical analysis for our GDTPP will be our future work.
> > >
> > > [1] Kashif Rasul, Calvin Seward, et al. Autoregressive Denoising Diffusion Models for Multivariate Probabilistic Time Series Forecasting, ICML2021.
> > > [2] Kashif Rasul, Abdul-Saboor Sheikh, et al. MULTIVARIATE PROBABILISTIC TIME SERIES FORECASTING VIA CONDITIONED NORMALIZING FLOWS, ICLR2021
> > >
> > > THANKS A LOT!

---

### Review · Reviewer_o8BL · 2022-06-08

**Summary Of Contributions:**

This paper revises the self-attentive encoders with adaptive reweighting for history encoders and adopts recently popular generative models, e.g. Conditional DDPM, VAE and others, to learn the temporal point processes.

**Requested Changes:**

1. In Line204, what is q(t^0)? Known Standard Gaussian, unknown distribution, or empirical distribution? It is not clear.
2. In Eq.12, I object to the assumption that q(t^k|t^{k-1}) is Gaussian. By definition, it is obvious that t^k> t^{k-1}, however in Gaussian assumption t^k \in R. Similar situations exist for p(t^k) and Eq.13.
3. For the original DDPM models, the interpolation between Gaussian noise and original data is one of the important reasons for good performance. However, similar data augmentation operations do not seem to be used.
4. The full name of CDD should be specified. And the names of the methods should be related to TPP to distinguish the original models.
5. The writing of contributions should be improved.

Minor comments:
1. Line79, exploring->explore
2. Table 1, Discription-> Description; exsiting-> existing
3. Line177, P.1. is not clear enough.
4. Figure.1 -> Fig.1 or Figure 1. Similar situations exist for others.

**Strengths And Weaknesses:**

1. The idea is clear and new.
2. The description of existing methods is clear in Table 1.
3. Experiments are clear.

---

> ### Author Response · Authors · 2022-06-09
> **Response to Reviewer o8BL with new submission**
>
> Thanks for your valuable advice. We have revised our paper according to your advice. Please read our responses together with the new version.
> 1. It is an unknown distribution which we aim to use $p_\theta(t)$ to approximate, we revised it as shown in Line. 268
> 2. $t^k$ is the k step latent variables in the diffusion model (Line. 211), not in the observed arrival time sequences, so $t^k$ is not necessarily larger than $t^{k-1}$. However, in our observed sequence, $t_i > t_{i-1}$, so we actually model the time interval $\tau_i =  t_i - t_{i-1}$ instead of timestamps $t_i$. We use the notation $\tau$ instead of $t$ for clarity in the newly-submitted version.
> 3. In the forward process, the noise is added. As you can see from Eq.(12). $\tau^k = \sqrt{1-\beta_k}\tau^{k-1} + \beta_k z$, which is a Markov chain with noise gradually added during the forward diffusion process. And in training,  as shown Eq. (14), the MSE between noisy augmented data and approximated data as variational bound is used for training. More specifically, in Line 277 and 278, $\tau'_i = \sqrt{\bar{\alpha}_k} {\tau}_i^0 + \sqrt{1 - \bar{\alpha}_k} {\epsilon}$ is the noisy-augmented input.  In our implementation code, the workflow is based on (Ho et al., 2020), you can check it in Line. 284 in `./models/prob_decoders/diffusion_modules/gaussian_diffusion.py'.
> 4. We add Temporal before each model, such as TCDDM, TCVAE, and so on.
>
> Other minor concerns are also revised. Thank you again for your detailed comments.

---

### Review · Reviewer_bbwf · 2022-06-28

**Summary Of Contributions:**

Temporal point processes (TPPs) are generative models for asynchronous event sequences occurring in continuous time.
Most recent papers on neural TPPs evaluate goodness-of-fit of these models using negative log-likelihood (NLL).
The authors of this work point out that good NLL scores may not be correlated with good predictive performance of TPP models (i.e., good sample quality).

The main contribution of this work aims to improve the sample quality by modifying the probabilistic **decoders** used in neural TPP models.
Specifically, the authors propose several new decoders for neural TPPs based on other modern deep generative model architectures
- diffusion model
- variational autoencoder
- generative adversarial network
- continuous-time normalizing flows.

These new architectures combined with the respective training procedures lead to better time prediction compared to existing NTPP architectures.

The second contribution of this work is a modification to the self-attention **encoder** architecture used in some neural TPPs.
The new modified parametrization of the self-attention layer
- replaces the positional embeddings with exponential smoothing and
- explicitly models dependencies between different mark types (similar to a low-rank Granger causality matrix)


**Broader Impact Concerns:**

There is no Broader Impact Statement provided, but I don't believe that there are any major ethical concerns that haven't been addressed.

**Requested Changes:**

The following points are critical for securing my recommendation for acceptance:

1. Clarifying my concerns regarding the proposed attention mechanism and its connection to the Granger causality matrix (see point #2 above).
2. A quantitative evaluation on the structure discovery task.
3. Comparing to a purely discriminative baseline that directly predicts the next inter-event time $\hat{\tau}$ (see point #4 above).

**Strengths And Weaknesses:**

Strengths:
- This paper highlights an important problem with existing neural TPP models ("good NLL does not imply good sample quality"). This is highly relevant to the practical applications of TPPs in predictive tasks.
- This work provides a thorough discussion of the different possible architectural choices in neural TPP models (both encoder, decoder and training procedures) and nicely connects to the previous works.

Weaknesses:
1. The code provided with the submission is incomplete. The `models` directory that should contain the implementations of all models is missing; and the link to Google Drive with the source code seems to be broken.
Because of this I couldn't resolve some of my concerns below by looking at the source code.

2. The introduced "relation between event types" mechanism (property P.2 / L184 / Eq. 11) seems to be incorrect.
    - According the Eq. 11 and Fig. 5, the weights $w_{j, i-1}$ can be negative since the event type embedding matrix $E_m$ is not guaranteed to be positive. Because of this, we cannot interpret the weights $w_{j, i-1}$ as attention coefficients anymore.
    The denominator in Eq. 11 can even be equal to zero or negative, breaking the attention mechanism.
    - Even if we restrict the mark embedding matrix $E_m$ to be positive, we still cannot interpret the matrix $W = E_m^T E_m^T \in \mathbb{R}^{M \times M}$
    in Eq. 11 as modeling interactions between marks / capturing Granger causality in the sense of https://arxiv.org/abs/1602.04511.

        For the correct definition of Granger causality, we need the following property. For each past event $j$ and any mark $k \in [M]$, it must hold that $W_{m_j, k} = 0 \implies$ "$p^*(t_i | m_i = k)$ is independent of event $(t_j, m_j)$".
        However, the definition provided in Eq. 11 uses the observed mark $m_{i-1}$ of the previous event number $i-1$, not the possible marks $k$ of the next event number $i$.
        Therefore, the matrix $W$ doesn't carry the usual intuitive meaning of the Granger causality matrix for TPPs.

3. Certain aspects of the experimental evaluation do not convincingly demonstrate the benefits of the proposed models.

    - The improvements in predictive performance of the new encoder architecture (Table 4) are really minor.

    - The learned similarity matrix (Figure 5) is quite dense, and it's not clear whether the larger values here indicate actual dependencies between event types or noise.
    See also the point #2 above.
    A quantitative evaluation compared to other methods for structure discovery (e.g., Hawkes process, [CAUSE](https://arxiv.org/abs/2002.07906)) would be more convincing.

    - It's not clear whether the improved performance is based on the differences between the models or peculiarities of the training procedure (such as early stopping).

        For example, all proposed model variants (TCDDM, TCVAE, TCGAN, TCCNF, TCNSN) as well as the baselines (RMTPP, LogNorm, WeibMix, FNNInt) use the exact same parametrization of the mark distribution $p^*(m_i)$ and are trained using the cross-entropy loss for marks.
        However, the accuracy scores achieved by different models in Table 3 are quite different, which is very suprising.

4. While the motivation for the proposed approach makes sense ("low NLL doesn't imply good samples"), it's not clear whether a much simpler baseline can't achieve the same or even better results.
For example, why don't we use a purely discriminative baseline that directly predicts the next inter-event time $\hat{\tau}$ and is trained by minimizing MAPE?
In my opinion, comparing to such a baseline is important if the focus of the paper is on inter-event time prediction.


Minor comments:
- Several points that need to be clarified in the paper:
    - Does the LogNorm baseline use a single log-normal distribution or a mixture, as in the original reference?
    - When computing mark prediction accuracy, do we consider $p^*(m_i)$, the marginal distribution of the next mark, or $p^*(m_i | t_i)$, the distribution of the next mark at time $t_i$?
    The two quantities are identical for the proposed models where the marks are independent of the arrival time (Sec. 3.4), but are different for the SAHP and TAHP baselines.
    - Do SAHP and THP models use the original encoders or the modified encoder (Eq. 11)?
- Some missing citations:
    - [Zhang et al., ICML 2020](https://arxiv.org/abs/2002.07906) also learn dependencies between event types in a marked neural TPP.
    - [Mehrasa et al., 2020](https://openreview.net/forum?id=rklJ2CEYPH) also use continuous-time normalizing flows / VAEs to model the inter-event time distribution in TPPs.
    - [Mei et al., ICLR 2022](https://openreview.net/forum?id=Rty5g9imm7H) also explicitly model dependencies between event types in the self-attention layer of a TPP (Eq. 8).
    - [Ben Taieb, AISTATS 2022](https://proceedings.mlr.press/v151/ben-taieb22a.html) also uses CRPS to evaluate TPP predictions.
- L88: Usually $N(t)$ is defined as the number of events in $(0, t]$, not $[0, t)$ (e.g., see in Daley & Vere-Jones).
- L155: The objective function of WGAN doesn't provide a lower bound on the likelihood, but is an approximation to the Wasserstein distance between the empirical and model distributions
- L369: It should be possible to sample events from SAHP and THP using Ogata's thinning method since the intensity for both methods is monotonically decreasing between events.
It's also possible to sample from the FNNInt model using numerical root-finding.

---

> ### Author Response · Authors · 2022-06-29
> **Response with new version updated**
>
> 1. Regarding the first point, I'm sorry that we accidentally left out the `models` folder when uploading the files. The updated supplementary file has now been uploaded.
>
> 2. Regarding equation 11, in fact, we should have written it as $w_{j,i-1} = \exp(({E}_m^T{m}_j)^T({E}_m^T{m}_{i-1})\exp{{a(t_{i-1}-t_j)\}}\phi({e}_j, {e}_{i-1})) /\sum_{j=1}^{i-1}\exp(({E}_m^T{m}_j)^T({E}_m^T{m}_{i-1}) \exp{{a(t_{i-1}-t_j)}}\phi({e}_j, {e}_{i-1}))$, so that the weight of attention would be positive. Since we need multiple exp() functions here, we did not write it that way. This corresponds to line 23 of /models/basic_layers/singleatt.py in our code. We have improved the version to make it easier for the reader to understand. See Eq.(10) and (11).
> Regarding the matrix, we do not treat it as granger causality, so this matrix is modeled only to demonstrate correlations learned by the model, not causal patterns of events in time. You can think of it as a graph structure between events, and our idea is consistent with <Learning Neural Point Processes with Latent Graphs> (https://dl.acm.org/doi/pdf/10.1145/3442381.3450135). Different from it, there are also positive and negative correlations, so we believe that the elements in this matrix can be negative.
> We were never meant to be a model that aims to recover correlations of events, and the structure learned by the model here is only for demonstration and not for evaluation or demonstrating its superiority. We hope that readers and reviewers will focus more on the theme of this article: Exploring Generative TPP, rather than Discovering the Granger Causality or Recovering Events' Relation Structure. Please give more attention to our generative models’ part, because our model is not built for relation or causality discovery. It is just a minor revision that we empirically prove to be useful.
>
> 3. The improvement from the encoder is indeed minimal, as demonstrated in EDTPP (https://arxiv.org/abs/2110.09823), and we also discuss it in Line.193. Because it is not our main contribution, the minor improvements are acceptable as we regard it as a small trick to further improve the expressivity of attention encoder.
> Regarding the correlation matrix, our analysis is in line with <Learning Neural Point Processes with Latent Graphs>, and we just give a simple demonstration. In fact, the analysis of correlation is not our main contribution and problem to be solved, and it takes up very little space in the paper. We only show experimentally that such simple improvements can show consistent interpretability without breaking the original performance, rather than the mean of weight in the attention mechanism in which the variance of the weight of two event types may be large.
> The difference in prediction results for the next event type is indeed difficult to explain, because it involves a complex mechanism within the network. The historical encoding obtained by the encoder is used to model both event type and time, and this complex coupling mechanism does not allow us to explore it deeply.
>
> 4. If the time of occurrence of the next event is directly modeled as a regression task, the deterministic model losses uncertainty, which is no longer a probabilistic model. There is also no meaning when comparing CRPS. Here, we did an experiment with an architecture of encoder followed by two heads: one for the prediction of the next event type and the other for the prediction of the next event time. We report the MAPE and ACC for comparison purposes, which are updated in Table 3 and Line. 369.
>
> Minor:
> 1. Lognorm and all the classical probabilistic models all use a mixture, with 16 lognorm/Gaussian/Weibull … used.
> 2. Here we model them all uniformly in the same architecture: $ p^*(m,t)=p^*(m) p^*(t)$.
> 3. In Sec.5.2, all the methods are evaluated with the encoder as the revised attention (transformer).
> 4. We describe the sampling of these methods as inflexible, which means the trivial sampling methods can not be used, and the original version of these works does not provide sampling methods. We added your comments to this description for readers to follow our work better.
> 5. Other points like citation and notation have been revised.

---

> > ### Comment · Reviewer_bbwf · 2022-07-18
> > **Response from the reviewer**
> >
> > Thank you for your response and for addressing my concerns regarding the definition of the attention weights $w_{j, i-1}$ and the model architectures.
> >
> > However, I still have some concerns regarding the encoder architecture / relations between event types (points 2 and 3 in your response).
> > Currently the revised encoder (with new positional encoding + explicit dependency matrix) is described as one of the 3 main contributions of the paper (line 76). It is stated that the new encoder provides improved **interpretability** and **expressiveness** (lines 76, 191, 201, 427, 451). However, in my opinion, these claims are not supported by the experiments or theory.
> > - It's not clear from the definition (Equation 12), how a human can interpret the learned matrix $W = E_m E_m^T$. As you mentioned, this matrix doesn't **directly** tell us anything about the occurrence of **future** events and doesn't capture Granger causality (so we cannot make conclusions like "$W_{k,l} > 0 \implies$ event of type $k$ in the past increases the probability of observing an event of type $l$ in the future"). Rather, the matrix determines how the previous event $i-1$ is combined with events $1, ..., i-2$ when computing the history embedding in the self-attention layer. This indeed captures some relation between the event types, but I don't understand how to interpret it.
> > - The learned matrix $E_m E_m^T$ shown in Figure 6 is quite dense, so it's not clear if it actually recovers some structure or if we're looking at noise.
> > - As you pointed out, the changes in predictive performance in Table 4 are quite small, and sometimes the standard deviation is as large as the observed improvement. We would need to re-run these experiments over many hyperparameter values / datasets to conclude that these indeed provide a consistent improvement.
> >
> > I see the value of contributions #1 and #3 of your paper, and think that it's overall an interesting and meaningful work. The only problem that I currently see are the claims about contribution #2 (the encoder) that are not well supported. I see two ways of fixing this part:
> > - More experiments on datasets with some known relations between event types (e.g., multivariate Hawkes processes) that clearly demonstrate that the matrix $E_m E_m^T$ captures these relations (qualitative or quantitative evaluation are both fine).
> > - Removing the part about the encoder from the paper, or at least adjusting the claims of interpretability and expressiveness.
> >
> > Apparently, there is some bug on OpenReview - I still cannot see the `models` folder in the supplementary materials, and the link to Google Drive still appears to be broken.
> >
> >
> > Some small remarks:
> > - It would be helpful to include the precise definition of function $\phi(\cdot, \cdot)$ to the paper.
> > - How did you modify SAHP and THP to make them conditionally independent? The original model definitions (Equations 15 and 6 in original papers) define a separate intensity for each mark, so do not permit a factorization  $p(t, m) = p(t) p(m)$.
> > - Even though the original code by Omi et al. doesn't include an implementation of the sampling method, it seems unfair to say that sampling for their approach is less convenient than, e.g., in score-matching models proposed in this work. Langevin dynamics for score-matching models require quite a bit more effort to implement compared to root finding that we can do with a few lines of code using [`scipy.optimizer.root_scalar`](https://docs.scipy.org/doc/scipy/reference/generated/scipy.optimize.root_scalar.html#scipy-optimize-root-scalar), something like
> >
> > ```python
> > def sample(lambda_integral):
> >     z = np.random.exponential()
> >
> >     def f(t):
> >         return lambda_integral(torch.as_tensor(t)).detach().numpy() - z
> >
> >     return scipy.optimizer.root_scalar(f, bracket=(0, 1000))
> > ```

---

> > > ### Author Response · Authors · 2022-07-19
> > > **Response to Reviewer**
> > >
> > > 1. Thanks for your advice. Our experimental protocol is consistent with <Learning Neural Point Processes with Latent Graphs> (https://dl.acm.org/doi/pdf/10.1145/3442381.3450135) and also, SAHP which gives attention matrix for interpretability, and the graph structure/ events' relation is unveiled in a totally unsupervised way. Therefore, we follow it and did not give further quantitative analysis on the relation. This matrix is used to get some insights into how the model captures the event's relation.
> > > We believe that our experiments and improvements are justified if there are no obvious flaws and the experimental protocol and results are acceptable in the previous works.
> > >
> > > 2. We upload our code again, and check it further. You can download it to get more details now :)
> > >
> > > 3. The precise formulation of the attention kernel is revised in the newly uploaded version.
> > >
> > > 4. It is the same as other methods, where two heads are stacked after the transformer: one to map the embedding into the continuous time-interval probability function, and the other to map it into a categorical probability distribution.
> > >
> > > 5. Thanks for your advice to solve the sampling problem in FNN. The method will be added to our framework in feature work.

---

> > > > ### Comment · Reviewer_bbwf · 2022-07-21
> > > > **Regarding the learned graph structure**
> > > >
> > > > Thank you for your answers, they address most of my concerns but I'm still not convinced by point #1. From the definition it follows that the attention scores $\phi(e_i, e_j)$ can be both positive and negative. This means that the **signs** in the matrix $W = E_m E_m^T$ do not carry a meaning --- a very negative value can both increase and decrease the influence of past events if the dot product $\phi(e_i, e_j)$ is negative or positive, respectively. Even if $W_{kl} = 0$, this can effectively **increase** the influence of past events of type other than $k$, if the other event types have a negative dot product with the reference event (so the pre-softmax weights may look like $(-5, -5, -5, 0, -5)$).
> > > >
> > > > I understand that "Learning Neural Point Processes with Latent Graphs" uses a similar evaluation strategy (however, in their case the influence matrix is strictly positive, so they don't suffer from the problem with the signs), but I'm now referring to the claim about interpretability of the GNTPP model in the paper under review (lines 78, 192, 195, 202, 428, 452). In my opinion, if a claim is made that a method is interpretable, then there should be some explanation about how I, as a human, can interpret it. In this case, I'm not sure how to interpret the values of the matrix $W$ because of the issues outlined above and in my previous comments (not related to Granger causality / future events; sign ambiguity). I will be happy to change my mind when presented with a clear explanation of how to interpret the weights, preferably supported by some empirical evidence.

---

> > > > > ### Author Response · Authors · 2022-07-22
> > > > > **Response**
> > > > >
> > > > > First, on your question about pre-softmax: In implementation, we first generated a mask (Line 44 in '/models/hist_encoders/attention.py'), and then we map the 0 to -1e9 (Line 20 in '/models/basic_layers/singleatt.py') to force the influence of past event to be 0 after the softmax. It is very important while we omit in our paper, which may cause confusion to readers. And we add the explanation in our newly-updated version.
> > > > >
> > > > > You are right on the claim of the sign of $E_m E_m^T$, and it is different from "Learning Neural Point Processes with Latent Graphs". Actually, we did not restrict it with positive constraints for a higher degree of freedom of network parameters. We hope to use cosine similarity to demonstrate their relationship, and to train it as an inductive bias in the attention mechanism, without disturbing the degree of the freedom of model parameters too much (in other words, help to improve the expressivity.)
> > > > >
> > > > > After our explanation of why it is not positive, If you insist that the positive value in $W$ can show more interpretability as in "Learning Neural Point Processes with Latent Graphs", we can restrict it with a positive constraint as $exp(W_{i,j})$ and update the experimental results.
> > > > >
> > > > > However, we think the evaluation in SAHP and "Learning Neural Point Processes with Latent Graphs" as demonstrating the learned event's relation is valid. Your advice on the empirical study on how to interpret the relations is insightful, but in this work, we follow these two papers' experimental protocols, and to a large extent, our work is much more complete than the previous works. More evaluation with quantitative metrics like some work in granger causality discovery will be our feature targets.
> > > > >
> > > > > We still hope reviewers and readers take more attention to the probabilistic decoder parts, and when they need to see the learned relation patterns, they can refer to the matrix $W$.
> > > > >
> > > > > Thanks!

---

> > > > > > ### Comment · Reviewer_bbwf · 2022-07-25
> > > > > > **Regarding the learned graph structure**
> > > > > >
> > > > > > Thank you for the response. I feel like we are really focusing on the details here, so I just want to again restate my main concern to make it clear to you and other readers. To be clear, I like the paper and I think that contributions 1 and 3 are important, interesting and supported by the experiments. The only problem I have is the claim in contribution 2 (interpretability) that seem to not be supported by theory and / or experiments.
> > > > > >
> > > > > > The paper repeatedly claims that the matrix $E_m E_m^T$ is interpretable but doesn't explain how it should be interpreted. What I'm looking for is something like following statement with a clear mathematical explanation (not just "$W_{kl} \ne 0$ means that there is some relationship between event types $k$ and $l$"):
> > > > > >
> > > > > > > This is how we can interpret the learned matrix $W = E_m E_m^T$
> > > > > > > - $W_{kl} = 0$ implies that ...
> > > > > > > - $W_{kl} < 0$ implies that ...
> > > > > > > - $W_{kl} > 0$ implies that ...
> > > > > >
> > > > > > For example, in case of a linear Hawkes process we can interpret the influence / Granger causality matrix as
> > > > > > - $W_{kl} = 0$ implies that $\lambda_l(t| \mathcal{H}_t)$ (conditional intensity of event type $l$) doesn't change if we remove events of type $k$ from the history $\mathcal{H}_t$.
> > > > > > - $W_{kl} > 0$ implies that $\lambda_l(t| \mathcal{H}_t)$ increases if we add an event of type $k$ to the history $\mathcal{H}_t$.
> > > > > >
> > > > > > As we discussed before, the matrix $W$ in the proposed model doesn't capture Granger causality, so the above explanation doesn't apply to GNTPP and another explanation would be needed to support the interpretability claim.
> > > > > >
> > > > > > ----
> > > > > > Now, regarding your explanation of the softmax and the updated paper.
> > > > > >
> > > > > > If I understand the code correctly, padding with -1e9 is done to enforce causal masking, not masking based on the embedding matrix $E_m$, though I might be wrong. The description in the paper (lines 207-210) looks a bit strange - we randomly initialize $E_m$ and optimize it with gradient descent, so it's extremely unlikely that we will ever get $(E^T_m m_j)^T (E^T_m m_{i-1})$ **exactly** equal to zero. Even if this happens, replacing the zero values with 1e-9 is a non-differentiable operation. Even if that wasn't the problem, the term $\phi(e_j, e_{i-1})$ can also be negative, so we will end up increasing the influence of event $j$ instead of ignoring it.

---

> > > > > > > ### Author Response · Authors · 2022-07-26
> > > > > > > **Response**
> > > > > > >
> > > > > > > Thank you for your recognition of our contributions.
> > > > > > > To map the $W_{kl} = 0$ to 1e-9 implies that the k-th event has no impact on the occurrence on l-th. However, as you demonstrate, $W_{kl} < 0$ and $W_{kl} > 0$ will all mean that the impact exists, while the order of it does not imply anything. In this way, we think that forcing all the elements to be non-negative may be a good choice for interpretation?  And that a small value such 1e-7 may be used such that $W_{kl} [W_{kl}<1e-7] =0$ will avoid the problem of non-exactly-equal-to-zero.
> > > > > > >
> > > > > > > However, if we add a threshold and think of $W_{kl}[-1e-7 <W_{kl} < 1e-7] = 0 $ as no influence, and others as influence exist, the demonstration of models' learning relation between events can also be valid, and the expressivity (degree of freedom of model parameters) of the encoder may not be destroyed without the positive constraint.
> > > > > > >
> > > > > > > The interpretability is indeed minor, as you implied, and we think that it is a bonus brought by the event embeddings. If you will change your mind when the $W_{kl}$ is all set positive, as in the first paragraph we showed, we can revise the encoder part and experiment part correspondingly in the final version, while it will take time to test the metric of new encoder again.

---

> > > > > > > > ### Comment · Reviewer_bbwf · 2022-07-26
> > > > > > > > **Response**
> > > > > > > >
> > > > > > > > The main reason I disagree with the statement "$W_{kl} = 0$ or $W_{kl} = -1e9$ implies that the $k$-th event type has no effect on the occurrence of event type $l$" is that we use the weight $W_{kl}$ when aggregating events $\\{1, ..., i-1\\}$ **after** the event $i-1$ of type $l$ already occurred. However, we will use the result of aggregation $h_{i-1}$ to predict event $i$ whose type we don't know yet. The type of the future event $i$ (that the model tries to predict) has no effect on which past events we ignore.
> > > > > > > >
> > > > > > > > This is very different from the usual definition of Granger causality / influences between event types in a marked TPP that captures the influence of past events on occurrence of **future unobserved** events.
> > > > > > > >
> > > > > > > >
> > > > > > > > Back to the technical details:
> > > > > > > > - According to Eq. 12, setting $W_{kl} = -1e9$ doesn't guarantee that we will give 0 after-softmax weight to past events of type $k$. If $\phi(e_j, e_{i-1}) < 0$, which can happen based on the definition, we will end up giving 1 after-softmax weight to the past event of type $k$.
> > > > > > > > - According to Figure 6, most entries of the influence matrix are < 1e-7 or > 1e-7, so it's unclear what conclusion we should draw from it other than most event types are related to most event types.

---

> > > > > > > > > ### Author Response · Authors · 2022-07-27
> > > > > > > > > **Response**
> > > > > > > > >
> > > > > > > > >  You are right that it does not agree with the granger causality because it is a relational similarity matrix. And most previous works are also focusing on the impact of previous event $m_j$ on the latest one $m_{i-1}$. For example, in SAHP, the expectation of attention matrix also shows the impact learned by attention mechanism of $m_j$ on $m_{i-1}$. As shown in Eq. 12, our formulation is consistent with it.
> > > > > > > > >
> > > > > > > > > Technical details:
> > > > > > > > > 1. Based on the definition, chances are that the learned $W_{kl}$ is extremely small, and $W_{kl} = -1e9$ may not force the value after softmax to be 0.  We agree with you on this point, but it is noted that in Line 206, we have constraint $E_m^Tm$ as a unit vector, to make the inner product equivalent to cosine similarity., therefore, $-1\leq W_{kl}\leq 1$, the problem is avoided.
> > > > > > > > > 2. We agree with your concern. And we just provide a scheme to avoid the problem of not-exactly-equal-to-zero. If you think the threshold to force elements in the influence matrix equaling zero can brings improvements to our formulation of encoders, we will revise our code and obtain new experimental results to update this part.

---

> > > > > > > > > > ### Author Response · Authors · 2022-07-27
> > > > > > > > > > **Response**
> > > > > > > > > >
> > > > > > > > > > We here are all discussing the encoder relational matrix. I can either reduce his importance in the context, which after all is not the main contribution of this article, or further improve its part. What is your opinion on this, and would it better assist the reader if I divided the contribution into `minor` and `major`, and then introduced them separately? Your comments are very professional, and we are more than willing to accept and adopt them.

---

### Review · Reviewer_C6d5 · 2022-07-01

**Summary Of Contributions:**

In this paper, the authors study several generative neural temporal point process models. For the history encoder part, the authors modify the self-attentive encoder by adding learnable time encoding and incorporating event type encoding. On the decoder part, various generative models are adopted, including diffusion models, VAE, GAN, normalizing flows, and score-based generative models. Finally, the authors conducted extensive experimental studies to evaluate the performance of different generative TPP models on several benchmark datasets.


**Requested Changes:**

* Provide more insights into the choice of different probabilistic encoders, their respective pros and cons.
* Consider alternative evaluation methods for generative models if applicable, quantitatively (e.g., comparing likelihood/ELBO) and qualitatively (e.g., latent variable analysis in https://arxiv.org/abs/1506.02216)


**Strengths And Weaknesses:**

Strengths:

The paper is clearly written. All major modern generative models are considered as decoders. The experimental study is extensive. It should be a very good first paper to read for someone new to this field.

Weaknesses:

My impression of this paper is that it seems like something between a survey paper and a technical contribution paper. The paper has its technical contributions, but it is somewhat limited, since this is not the first paper applying generative models to TPP modeling. On the other hand, it is also not as systematic and comprehensive as a full fledged survey paper. For example, when comparing different generative models as decoders, there are not many insights on the pros and cons of each model, other than the empirical evaluation.

Coming back to the technical contribution part, one claimed contribution is the augmentation of the self-attentive encoder. I think the two problems listed in section 3.1 are definitely important considerations when designing the model. But I would argue that they are not necessarily “problems”, they are more like different modeling assumptions or inductive biases. Incorporating more features or learnable parameters will improve the performance of the model in some cases, but probably not on all datasets.

In my opinion, the advantages of generative models over discriminative ones are that (1) there is a quantitative evaluation of the goodness-of-fit, for example, the log-likelihood or ELBO; (2) the probabilistic model can be used for sampling or posterior inference. In the experimental study, the evaluation metrics seems to be focusing on the predictive quality of the models, rather than the generative capabilities. I would suggest adding more model comparisons from this perspective.

---

> ### Author Response · Authors · 2022-07-03
> **Response with new version updated**
>
> |                |     MOOC        |     Retweet     |     SOF         |     Yelp        |
> |----------------|-----------------|-----------------|-----------------|-----------------|
> |     RMTPP      |     1.7504      |     -1.9872     |     4.9031      |     -1.0832     |
> |     LOGNORM    |     1.3635      |     -2.4197     |     4.8782      |     -1.2808     |
> |     ERTPP      |     3.5791      |     -0.8876     |     5.0845      |     -0.9678     |
> |     WEIBMIX    |     0.7950      |     -2.5110     |     3.8717      |     -1.1125     |
> |     FNNINT     |     -2.3024     |     -3.0064     |     1.8469      |     -1.2294     |
> |     SAHP       |     -2.3472     |     -2.9955     |     1.8348      |     -1.6607     |
> |     THP        |     0.1270      |     -1.3794     |     1.8591      |     -1.6349     |
> |     TCDDM      |     ≤1.7609     |     ≤0.7560     |     ≤2.2450     |     ≤0.0142     |
> |     TCVAE      |     ≤9.4754     |     ≤7.5911     |      ≤3.9727    |     ≤6.2181     |
> |     TCGAN      |     (0.0056)    |     (0.0001)    |     (0.0511)    |     (0.0560)    |
> |     TCCNF      |     -2.8591      |     -3.0464     |     1.6881      |     -1.8791     |
> |     TCNSN      |     ≤2.1815     |     ≤0.8340     |     ≤2.4131     |     ≤0.3517     |
>
> Thanks for your constructive advice.
>
> 1. We do follow part of the work: EDTPP, which is a comprehensive review of the Temporal point process, while some of the notations and terms are revised to fit in. However, we did not think of our work as a survey paper. The technical contribution is significant as we establish a systematic framework for the generative temporal point process.
> 2. We give the negative ELBO which is the upper bound of models’ NLL of TCDDM, TCVAE, and TCNSN, and the exact NLL of other models except that in TCGAN we give the Wasserstein distance between the empirical and model distributions. It is added in the supplementary. Here we claim that the CRPS can also be regarded as an alternative of NLL to evaluate the model’s fitting ability, which has been widely used in time series and temporal point processes [1] [2] [3].
> 3. A deep insight of analysis of the sampling dynamics of TCDDM, TCNSN, and TCCNF has been conducted in our new version, as shown in Line. 407 to 425 and Figure 5, which can be regarded as the flow of embedding in the neural network, and an explanation of why TCNSN fails to model TPP from the perspective of stochastic differential equation is given. We hope it can give more insights into how these models work in TPP. Please refer to our new version in detail.
>
> [1] Kashif Rasul, Calvin Seward, et al. Autoregressive Denoising Diffusion Models for Multivariate Probabilistic Time Series Forecasting, ICML2021.
>
> [2] Kashif Rasul, Abdul-Saboor Sheikh, et al. MULTIVARIATE PROBABILISTIC TIME SERIES FORECASTING VIA CONDITIONED NORMALIZING FLOWS, ICLR2021
>
> [3] Souhaib Ben Taieb. Learning quantile functions for temporal point processes with recurrent neural splines. In Gustau Camps-Valls, Francisco J. R. Ruiz, and Isabel Valera (eds.), Proceedings of The 25th International Conference on Artificial Intelligence and Statistics, volume 151 of Proceedings of Machine Learning Research,
> pp. 3219–3241. PMLR, 28–30 Mar 2022

---

### Decision · Action_Editors · 2022-08-01

**Recommendation:** Accept with minor revision

**Comment:**

The paper studies neural temporal point process (TPP) models, and makes the following two contributions:

- It explores a range of modern generative models for the TPP decoder.
- It proposes a modification in the attention mechanism of certain TPP encoders.

The reviewers found both contributions interesting and sufficiently evaluated.

However, as part of the second contribution, the paper contains the following interpretability claim (lines 76--78):

"we revise the self-attentive encoders with adaptive reweighting terms [...] showing better interpretability"

The reviewers were not convinced that this claim was supported by evidence.

I'm willing to recommend the paper for acceptance as long as the authors revise their manuscript to *completely remove all claims* that their proposed modification leads to improved interpretability. At a minimum, I would expect to see the following changes:

- Lines 77--78: delete "showing better interpretability".
- Line 202: delete "To provide interpretation of the model".
- Lines 431--432: delete "prove the better [...] interpretability of our revised attentive encoder".
- Lines 454--455: delete "experimental results show good [...] interpretability of our revised attentive encoders"